# `Data-Juicer` 2.0: Cloud-Scale Adaptive Data Processing for and with Foundation Models

**Daoyuan Chen**[*], **Yilun Huang**[*], **Xuchen Pan**[†], **Nana Jiang**[†], **Haibin Wang**[†]
**Yilei Zhang**[†], **Ce Ge**[†], **Yushuo Chen, Wenhao Zhang, Zhijian Ma**
**Jun Huang, Wei Lin, Yaliang Li**[‡], **Bolin Ding**[‡], **Jingren Zhou**
Alibaba Group

## Abstract

Foundation models demand advanced data processing for their vast, multimodal datasets. However, traditional frameworks struggle with the unique complexities of multimodal data. In response, we present `Data-Juicer` 2.0, a data processing system backed by 100+ data processing operators spanning text, image, video, and audio modalities, supporting more critical tasks including data analysis, synthesis, annotation, and foundation model post-training. With seamless compatibility and dedicated optimization for popular dataset hubs like Hugging Face and computing engines like Ray, it improves upon its predecessor in terms of usability, efficiency, and programmability. It features an easily accessible user interface layer that supports decoupled Python interactions, RESTful APIs, and conversational commands. Its new runtime layer offers adaptive execution across diverse scales and environments, abstracting away system complexities. Extensive empirical evaluations demonstrate `Data-Juicer` 2.0's remarkable performance and scalability, highlighting its capability to efficiently process TB-level data with 10k+ CPU cores. The system is publicly available and has been widely adopted in diverse research fields and real-world products such as Alibaba Cloud PAI. We actively maintain the system and share practical insights to foster research and applications of next-generation foundation models.

## 1 Introduction

**Data Processing Challenges for Foundation Models.** Foundation models require sophisticated pipelines for multimodal data across evolving paradigms in pre-training and post-training. While existing frameworks address specific aspects of text processing [18, 50] or traditional big data workloads [66], they lack essential capabilities for contemporary multimodal workflows. Three critical gaps emerge:

*Multimodal Processing Limitations:* Current tools provide inadequate support for cross-modal alignment and semantic-aware transformations crucial for vision-language models [34]. The transition from text-only systems like the inaugural version, `Data-Juicer` 1.0. [1] to multimodal processing introduces architectural challenges in handling heterogeneous data types and inter-modal relationships.

*Efficiency-Scalability Tradeoffs:* Traditional big data engines [52] and Python-based solutions [32] struggle with foundation models' unique computational patterns - simple per-sample operations

---

[*]Co-first authors.

[†]Equal contribution.

[‡]Corresponding authors, email addresses: {yaliang.li, bolin.ding}@alibaba-inc.com

[1]In this paper, references to `Data-Juicer` 1.0 pertain to the SIGMOD publication [8] and the open-source codes versioned with the v0.1.2.

39th Conference on Neural Information Processing Systems (NeurIPS 2025) Track on Datasets and Benchmarks.

applied at petabyte scales. This creates reliability risks in large-scale processing scenarios where late-stage errors can invalidate terabytes of computation.

*Ecosystem Fragmentation:* Disjoint APIs across popular frameworks force practitioners into suboptimal workflow choices. The lack of unified abstractions hampers portable optimization and cross-platform execution.

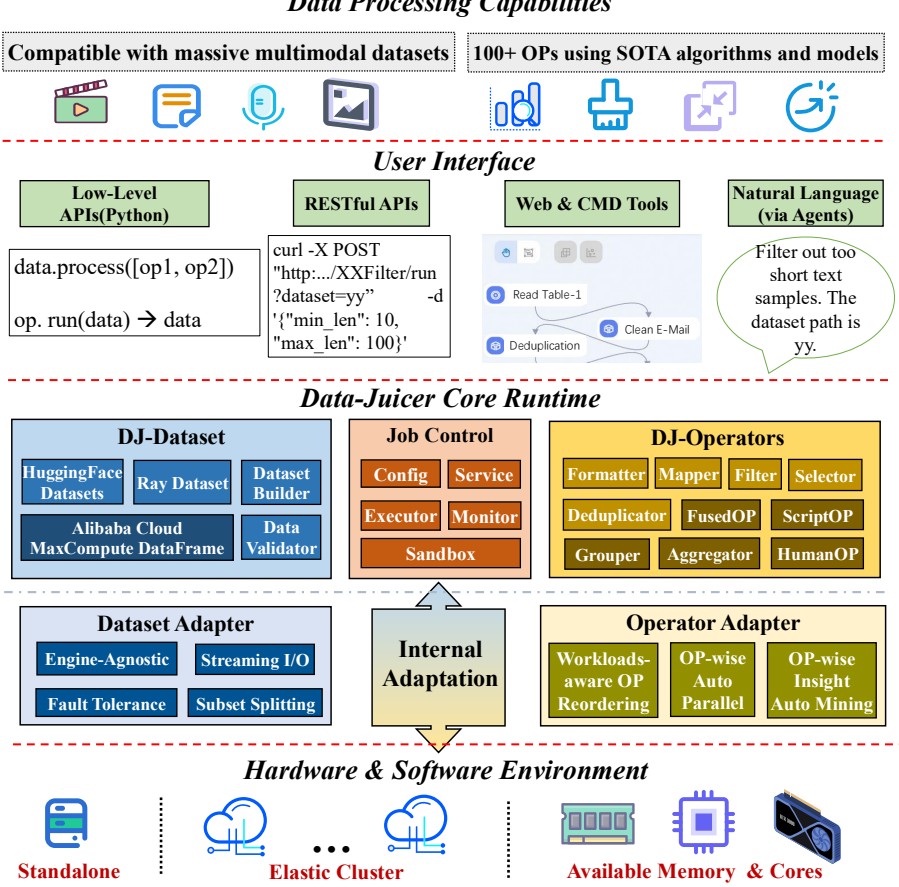

Figure 1: The overview of `Data-Juicer` 2.0.

**Architecture Overview.** `Data-Juicer` 2.0 addresses these challenges with a layered architecture optimized for foundation models (Fig. 1):

*Capability Layer:* Extends `Data-Juicer` 1.0's 50 text-only operators (OPs) for pre-training to 150+ multimodal OPs supporting text/image/video/audio processing and more post-training tasks. New OPs integrate foundation models (e.g., Tongyi-Qwen [51], SDXL [45]) for semantic-aware filtering and cross-modal synthesis while maintaining compatibility with mainstream dataset hubs [58, 59].

*Interface Layer:* Provides multi-level APIs balancing flexibility and accessibility. Low-level Python APIs enable custom pipelines, while RESTful endpoints and a visual editor support rapid prototyping. Novel agent integration allows natural language specification of pipelines.

*Runtime Layer:* Introduces four key enhancements: (1) A unified `Data-Juicer-Dataset` abstraction spanning local to cloud-scale execution environments; (2) Decoupled OPs with automatic optimization through adaptive batching and resource allocation; (3) Control plane supporting fault-tolerant execution and data-model co-development [9] for insight mining; and (4) `Adapter` components enabling automatic hardware-software configuration and parallelism across diverse deployments.

**Contributions.** Our contributions are summarized below:

*Enhanced System and Adaptive Techniques.* Incorporating feedback from numerous users and cutting-edge applications built upon its predecessor, `Data-Juicer` 2.0 emerges as a new open-source

system with enhanced multimodal processing capabilities. It emphasizes dedicated multi-layered adaptability and system optimization, integrating cloud-scale practices and diverse computation engines to dynamically and efficiently meet data processing demands.

*Extensive Evaluation and Usages.* We conduct thorough experimental analyses to assess the system's data processing capabilities under varied workloads, such as filtering-intensive, model-based, and semantically editable multimodal scenarios, using datasets ranging from millions to tens of billions of samples. We provide actionable insights and performance trade-offs across different use cases and resources, involving Ray and Alibaba MaxCompute [13] with up to 100 nodes and 12,800 cores.

*Community and Applications.* We have open-sourced this new system at `https://github.com/modelscope/data-juicer`, fostering sustained maintenance and engagement through practical events [47, 57], such as tutorials, surveys, data competitions, and co-optimization with community like Apache Arrow, Ray, and NVIDIA-Nemo-Curator teams. `Data-Juicer` 2.0 also facilitates many foundation model researches such as those from Alibaba Tongyi [25, 46, 2, 28, 35, 64, 69, 29], and serves as the operator base for multiple Alibaba Cloud products such as PAI Designer [14] and DLC-Ray [17], indirectly benefiting hundreds of internal and external customers across various real-world AI businesses. One of them has been running stably for over five months, processing data at a scale exceeding terabytes.

## 2 Preliminaries and Design Rationale

### 2.1 Related Work & Core Challenges

While existing systems address big data [66] or text-centric model data [18, 50, 27], a dedicated, open-source framework for multimodal foundation models is lacking. This gap stems from three core challenges rooted in the unique demands of modern AI workflows [47, 3]: **1) Functionality:** Processing requires deep semantic understanding and cross-modal alignment (e.g., for video, image, text, audio) [36, 34], moving beyond the structured data focus of traditional systems. **2) Efficiency:** Workloads are dominated by simple, per-sample operations (mappers, filters) at massive scale, often involving costly model inference, which contrasts with the complex aggregation queries optimized by conventional engines [66, 52]. **3) Usability:** The practitioner ecosystem is centered on Python, demanding native, intuitive interfaces that abstract away backend complexities [42, 32]. These challenges require new system designs, motivating the specific goals of `Data-Juicer` 2.0.

### 2.2 System Design Goals in `Data-Juicer` 2.0

**Comprehensive Multimodal Processing.** To address functionality gaps, extensive operators are required for collecting, cleansing, annotating, and synthesizing data across modalities like video, image, text, and audio, integrating both perceptual and cognitive information [7, 45].

**Efficient and Optimized Data Flow.** To tackle efficiency issues, we aim to accelerate high-frequency basic operators and efficiently manage high-cost semantic operations, while minimizing I/O and data transfer overhead.

**Intuitive and Engine-Agnostic Interface.** To enhance usability, we aim to protect users from the complexities of underlying execution engines, with easy-to-use Python APIs, graphical interface, and natural language interaction.

**Adaptive and Scalable Execution.** The system must adapt to diverse computational environments and workloads. It is designed to intelligently orchestrate and optimize data processing across various backends—from local execution to distributed computation on large-scale clusters.

### 2.3 Key Differences from Prior Systems

While its predecessor, `Data-Juicer` 1.0, laid a foundation for text data processing, it faced limitations in multimodal support, programmability, and fragmented workflows across different backends. `Data-Juicer` 2.0 is architected to overcome these issues. Unlike modality-specific toolkits [27, 43] or general big data systems [66], `Data-Juicer` 2.0 introduces a composable operator system that generalizes across data types and training tasks while retaining scalability. It distinctively integrates deep learning models as first-class citizens in the data pipeline, emphasizing both computational

Table 1: Comparison of Data-Juicer 1.0 and 2.0.

| Feature | Data-Juicer 1.0 | New in Data-Juicer 2.0 |
|---|---|---|
| Data Modality | Text-only (~50 OPs) | + ~100 Image, video, audio OPs |
| Major OP Types | Formatter, Filter, Mapper | + Grouper, Aggregator, FusedOP, HumanOP |
| Deduplicator | Standalone-only | + Ray-based distributed deduplication |
| Interaction | CLI, Low-level APIs | + RESTful APIs, Web UI, NL Interface |
| Execution Engines | HF Datasets, Beam | + Tighter integration with Alibaba PAI, Ray |
| Optimization | Greedy OP fusion | + Advanced OP reordering, auto resource allocation, batched/parallel execution, etc. |
| Compute Scale | 1,000+ cores | 10,000+ cores |
| Data Scale | ~70M samples (TB-level) | ~70B samples (PB-level) |
| Code Contribution | 26 PRs, ~34k LoC | + 100+ new PRs, ~40k added, ~8k deleted |

efficiency and semantic richness. Table 1 summarizes the key advancements from `Data-Juicer` 1.0 to `Data-Juicer` 2.0.

## 3 Processing Capabilities Beyond Text & Pretraining

`Data-Juicer` 2.0 extends its predecessor to support the processing of multimodal datasets on more tasks for a wide range of foundation models with about 100 new OPs. Images, videos, and audios are all valid inputs of `Data-Juicer` 2.0, and it's able to process datasets for post-training tasks, including supervised fine-tuning (SFT), Reinforcement fine-tuning (RFT), and so on. Detailed list of these OPs is in Appendix A. For clarity and ease of reference, the OPs are categorized from the following aspects, and their distributions are visually displayed in Fig. 2:

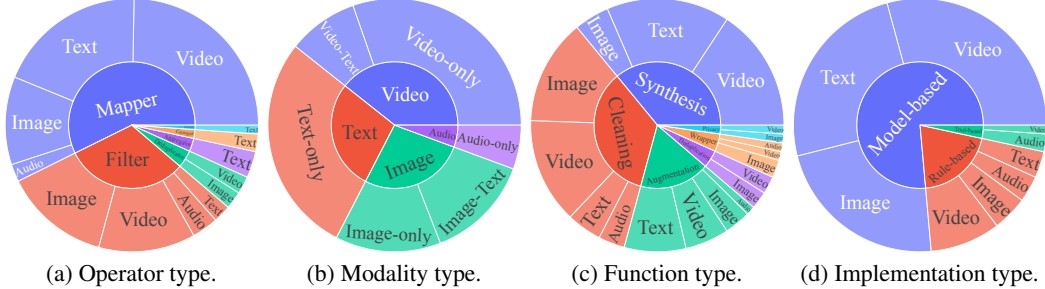

(a) Operator type.  (b) Modality type.  (c) Function type.  (d) Implementation type.

Figure 2: Distribution of new OPs across various dimensions. The high-resolution versions of these subfigures can be found in Appendix, Figure 11, 12, 13, and 14.

**Operator Types:** These new OPs are built on both original types in `Data-Juicer` 1.0 and types newly introduced by `Data-Juicer` 2.0. Among them, Formatters load datasets; Mappers edit samples; Filters compute data stats and remove samples accordingly; Deduplicators find redundant samples; Selectors sample data based on preferred ranks or rules; Groupers batch samples, and then Aggregators combine them into one. Notably, `Data-Juicer` 2.0 introduces a new type of OP named HumanOP. It's built upon Label Studio [61] and involves asynchronous human annotations and feedback during data processing and time-delayed downstream training tasks. This new OP helps to build SOTA human-in-the-loop procedures, such as reinforcement learning from human feedback (RLHF). These OP types will be detailed later in Sec. 5.2. Statistically, about 90% of OPs are Mappers and Filters, reflecting trends of user needs in the foundation model domain. New variants in other types, such as Formatters and Deduplicators, remain stable due to the broad applicability of already supported formats (e.g., JSONL, Parquet, MP4) and classical algorithms like MinHash [5].

**Modality Types:** The majority of new OPs are concentrated on video/image/audio/text-only processing, with about 20 OPs dedicated to cross-modal data processing. Among these new OPs, almost 3/4 of them are aimed at multimodal data processing, covering images, videos, and audio. There are both

single-modality OPs, such as `video_motion_score_filter` that scores the dynamics of videos, and cross-modal OPs such as `phrase_grounding_recall_filter` and `video_captioning_from_summarizer_mapper`, which measure the alignment between different modalities and generate contents from one modality to another, respectively. Detailed showcases are in Appendix B.

**Function Types:** Data cleaning and analysis operations constitute nearly 1/3 of the new OP suite. In particular, `Data-Juicer` 2.0 introduces about 50 OPs for data synthesis and augmentation on cross-modal, post-training, and reinforcement learning scenarios, enabling any-to-any generation among 4 supported modalities and improving textual dialogs based on varied information and needs. Furthermore, new privacy protection OPs, such as removing not-safe-for-work (NSFW) contents or blurring human faces, and several wrapper OPs for established 3rd-party tools, such as FFmpeg [21], have been integrated, enabling users to conveniently invoke existing professional commands.

Besides, 25 text-only OPs are introduced for post-training tasks. We tag them with fine-grained function types and summarize their distribution in Fig. 3. Among them, there are information extraction and synthesis OPs for curating dialog samples, as well as calibration and refinement OPs for optimizing questions and responses in SFT and RFT tasks. Beyond previous rule-based OPs to assess quality, difficulty, and diversity of datasets in `Data-Juicer` 1.0, more foundation-models-based OPs emerge to score the datasets from diverse dimensions and analyze rationale of given scores. This practice is flexibly and useful for vertical and cross-task scenarios. For example, rule-based methods usually struggle to analyze difficulties of math and financial problems, while this kind of OPs generally perform better with reduced customization and tuning efforts.

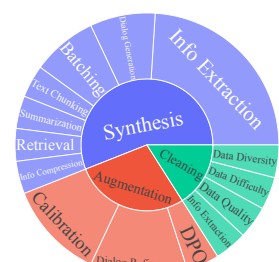

Figure 3: Distribution of fine-grained function types for text-only OPs. A high-resolution version can be found in Appendix, Figure 15.

**Implementation Types:** The new OPs include both novel algorithms from `Data-Juicer` 2.0 and implementations based on SOTA methodologies [49, 70, 22] from the community. Some OPs offer multiple versions to accommodate varying computational resources, such as CPU-only (e.g., with OpenCV [4]) or GPU-based configurations (e.g., with RAFT [60]). In the distribution, model-based OPs dominate, as semantic-aware processing often requires advanced models, such as large models for general-purpose understanding and generation via SDXL, GPT, and Qwen. Data processing with foundation models is becoming more popular and helpful.

## 4  Towards a More Accessible System

`Data-Juicer` 2.0 significantly improves accessibility over `Data-Juicer` 1.0 with multiple new interfaces catering to both novice and expert users.

In terms of user interfaces, in addition to the way to process data with an all-in-one configuration file provided in the previous version, we introduce 4 more user interaction methods. (1) **Low-level APIs.** We make the underlying implementation more transparent, and expose many extensible interfaces of `Data-Juicer-Dataset`, `Data-Juicer-Operators`, such that users can conveniently integrate `Data-Juicer` in their code. (2) **RESTful APIs.** For some web server usages, we provide one-click generation of high-performance web APIs capable of automatically discovering, registering, and adapting OP classes and tools. Users can start the server easily and trigger data processing across computing nodes. (3) **Web tools.** Based on RESTful APIs, `Data-Juicer`'s capabilities are integrated into Alibaba Cloud's visual modeling product, PAI-Designer [14], which provides a drag-and-drop UI for users to organize the data processing pipelines and allows users to make use of more products in the Alibaba Cloud ecosystem. (4) **Natural language interaction.** It's challenging to know when and how to use each of the 100+ diverse OPs. In response, we adopted AgentScope [24], a multi-agent platform, for low-code integration, using prompt-optimized reasoning and acting (ReAct) [65] agents to align OP functionalities with our RESTful APIs. Users only need to tell the agents how they want to process the dataset, and the system will automatically handle the job analysis and execution. More details about these interfaces are in Appendix C.

As the 100+ OPs in `Data-Juicer` 2.0 cover quite diverse multimodal data and training tasks, the complete capabilities require heavy dependencies and take the risk of runtime errors. This may slow down the system installation, initialization, and troubleshooting. We thus prepare a minimal set of requirements and split OPs' dependencies into subgroups based on their modalities and

usage categories. Users only need to install the lite `Data-Juicer` in a quick way, and the optional dependencies will be lazy-loaded when using specific OPs. Moreover, we maintain an automated unit and regression testing mechanism, ensuring over 85% test coverage. These actions make our system more user-friendly, helping both beginners and light users get started with it.

## 5 Towards a More Flexible, Robust, & Efficient Runtime

### 5.1 `Data-Juicer`-Dataset

**Unified Execution Abstraction.** `Data-Juicer` 2.0 introduces a `Data-Juicer-Dataset` class that abstracts heterogeneous computational engines (Hugging Face Dataset, Ray Data, MaxFrame-DataFrame), while preserving native API compatibility through a Facade-pattern design [23]. It provides unified interfaces for standalone/distributed execution modes, enabling transitions between environments while hiding engine-specific complexities. The class supports chainable processing workflows via templated methods with multiple `process()` call manners. More implementation details about these modes are provided in Appendix D.1 and Listing 2.

**Reliable Data Loading.** Data loading is now more systematic, moving beyond simple path specifications. The new `DatasetBuilder` class supports explicit dataset source specification (e.g., local, remote) and customizable configuration for loading datasets. To improve data loading reliability, a novel `DataValidator` module is introduced to check and validate data sources, schemas, and whether the dataset meets the processing goal. For example, post-tuning data should contain dialogs, and image paths should exist in image-captioning datasets.

**Token-Aligned Data Schema.** Our intermediate schema represents multimodal data through special tokens (e.g., `<__dj__image>`, where "dj" stands for `Data-Juicer`) in text fields, with chunk-based alignment using `<|__dj__eoc|>` separators. This token-centric design supports both simple cross-modal pairs and complex interleaved datasets (MMC4-style [70]), while preserving positional relationships and reducing redundant computing of shared media files. Core fields ("text", "query"/"response") also align with the popular formats of language foundation models, supplemented by bi-directional conversion tools for training ecosystems like LLaMA-Factory [68] and ModelScope-SWIFT [67] (implementation details are in Appendix D.2).

**Internal Adaptation with Industrial-Grade Optimization.** The intermediate layer design enables us to adaptively take advantage of different underlying engines and systematically improve usability. We can optimize once and apply the enhancements with both execution modes. For example, our minhash-based Deduplicators achieve engine transparency and performance superiority, yielding 3.3x speedup over vanilla Ray with load-balanced union-find [30] and hash-based aggregation.

Moreover, large-scale data processing faces errors from data corruption or unreliable operators (e.g., malformed LLM outputs), which can halt entire pipelines. `Data-Juicer` 2.0 enhances resilience at multiple levels. For operator-specific failures, it performs pre-flight validation on LLM responses and employs automatic retries with backoff. More systematically, to overcome the coarse-grained fault tolerance of underlying engines like Ray (which require full restarts), we introduce operator-level checkpointing and fine-grained recovery. This allows pipelines to resume from the last successful stage rather than from scratch, ensuring robust progress even with intermittent task failures. Schema-compatible empty samples are also used to maintain dataset integrity (e.g., see Fig. 7 in Appendix E.2). These optimizations are detailed in Appendix E.

### 5.2 `Data-Juicer`-Operators

**Composable Processing Primitives.** We extend `Data-Juicer` 1.0's five atomic OPs (Formatter/Filter/Mapper/Deduplicator/Selector) with five compositional types: Grouper/Aggregator/FusedOP/ScriptOP/HumanOP. Using Strategy/Decorator patterns [23], they enable flexible algorithm encapsulation and runtime behavior extension (OP taxonomy in Appendix D.3.1). FusedOP optimizes batch-wise OP fusion (Appendix D.3.2), while ScriptOP integrates custom Python logic. The Grouper takes a `Data-Juicer-Dataset` as input and groups data samples into batches, which can then be input into the Aggregator for subsequent aggregation. All OPs follow a unified template method `run()` with automatic parallelism configuration, decoupling execution logic from runtime engines.

We design an abstract factory class that centralizes common functionalities (parameter preparation, serialization) while allowing for implementation of OP-specific execution logic through overridable methods (e.g., `compute_stats()` for Filters). This eliminates executor dependencies and enables easy standalone customization/inspection/testing of individual OPs as their functionalists are constrained to be implemented in a self-contained manner.

**OP-wise Optimization.** We incorporate several automatic adaptation features for `Data-Juicer-Operator`, aiming at balancing resource constraints and operational efficiency without requiring users to understand hardware specifics or implementation details.

We employ a dedicated `Adapter` class, which uses a `probe_small_batch()` method to systematically probe and analyze essential information by applying individual OPs on randomly sampled data in runtime. As a result, we enhance `Data-Juicer` 1.0 's greedy OP fusion with adaptive reordering based on estimated OP speeds. Faster OPs precede slower ones within commutativity constraints, optimizing end-to-end latency (validation in Appendix H.2.1). Moreover, using a uniform parallelism granularity across all OPs in a data pipeline can cause OOM issues for some and resource underutilization for others. In `Data-Juicer` 2.0, with auto-configuration, model-based OPs use GPU/quantization (e.g., vLLM), while I/O-bound OPs use hierarchical parallelism across batched processing, multiprocessing and multithreading, taking the concurrent opportunities between I/O and computation latencies. The implementation details for OP `Adapter` are in Appendix F.

**OP Insight Mining.** The combined effects of sequential OPs are not always additive, as validated in [9]. Existing tools like `Data-Juicer` 1.0 [8] and Falcon [44] focus on coarse-grained metrics (e.g., data volume changes via Sankey diagrams) but lack fine-grained analysis. To formulate better data recipes, `Data-Juicer` 2.0 tracks dataset statistics (e.g., perplexity) and semantic tags (e.g., image categories) after each OP execution. Built-in `Analyzer` leverages Filters and modality-specific tagging OPs to generate histograms of statistical distributions and semantic categories (an example is shown in Appendix, Fig. 8). `Data-Juicer` 2.0 automates metric comparison between consecutive OPs, producing reports that highlight significant lineage-level variations. For example, a sudden text-length reduction after applying a BLIP-2 [33] image-text matching Filter could indicate noisy captions requiring adjustment. These insights help users systematically evaluate OP impacts, optimize data recipes, and identify unintended correlations.

### 5.3 Processing Job Control

**End-to-End Workflow Orchestration.** Our `Executor` module integrates configurable data pipelines with monitoring/checkpointing capabilities, codified into reusable data recipes (the red box in Fig. 1). We also provide a sandbox suite that enables data-model co-development through template workflows connecting to model training/evaluation infrastructures [9], allowing cost-effective exploration of data-compute effects before full deployment. More implementation details are in Appendix D.4. processing solution generation by foundation-model-based agents.

**Extensibility for Diverse Applications.** Many effect-proven and illustrative workflows have been encapsulated in YAML recipes and maintained online [53], catering to various vertical domains, such as multimodal data synthesis and persona-oriented dialog processing [28, 69, 54]. These facilitate interface exposure and reuse across different levels (Sec. 4), simplifying recipe routing and tailored processing solution generation by foundation-model-based agents.

## 6 Experiments & Insights

### 6.1 Experiment Setup

We evaluate `Data-Juicer` 2.0's efficacy across three data scales: small (560K-2.24M samples), medium (5.6M-56M samples), and large (56M-70B samples), covering both multimodal and text-only processing. Our test suite executes five representative data operations per recipe across three compute engines (Standalone, Ray, MaxCompute) using Alibaba Cloud resources (1-100 nodes, 64-12,800 CPU cores). All worker nodes maintain identical hardware configurations for fair comparison. More implementation details are in Appendix H.1.

## 6.2 Overview of System Performance Gains

**Macro-level Scalability** We systematically evaluated end-to-end performance by scaling datasets from 1x to 12,500x. The results, presented later from §6.3 to §6.5, confirm the robust scalability of `Data-Juicer` and provide actionable insights for choosing the optimal compute engine across different operational scales.

**Micro-level Optimizations** The strong end-to-end performance is underpinned by a suite of targeted optimizations. Below, we highlight the most impactful ones and point to where their effectiveness is validated (more details are in presented in Appendix H.2):

- **Resource Utilization & Adaptive Splitting:** Our adaptive data splitting for Ray offers a 2x-3x acceleration on large datasets. The mechanism's impact on reducing network I/O and improving CPU consistency is visually analyzed in Fig. 4f and discussed in §6.5.
- **Workload-aware OP Reordering:** For complex recipes, this strategy, along with OP fusion, can cut processing time by up to 70.22%. The quantitative benefits are detailed in the ablation studies in Fig. 9.
- **Automatic GPU Resource Allocation:** Critical for multimodal workloads, this prevents OOM errors and can save up to 99% of processing time. Its performance across various VRAMs is quantified in Table 4.
- **Batched Data Processing:** Optimizing batch size and concurrency is key. As shown in Fig. 10a, this can reduce processing time by up to 84%.

A key takeaway is that adaptive mechanisms for batching, resource allocation, and execution planning are crucial for mitigating hardware underutilization in modern data-centric AI pipelines.

## 6.3 The Case of Small Scales

**Performance Profile.** As shown in Fig. 4d, when processing small-scale multimodal datasets, the standalone mode with Hugging Face Dataset is efficient and comparable to the Ray mode with a single node. Additional Ray nodes provide further but limited acceleration (speedup ratios between 138% and 226% with 4 nodes). For text-only datasets (Fig. 4a), the standalone mode remains efficient. However, for the Ray mode, 4x node increments yield smaller speedups (148%) or even increased processing times due to dominated I/O and communication costs. Thus, *with `Data-Juicer`, processing datasets with hundreds of thousands of samples on a single machine is both efficient and cost-effective for most users*.

**Implications & Typical Application Scenarios.** In this scale, `Data-Juicer` enables two critical capabilities for data-centric AI research: (1). *Rapid recipe prototyping*. It simplifies data-model co-design as exemplified by extensive sandbox experiments [9], covering text-to-video generation, image-text pre-training, image-to-text generation for general image understanding, image captioning, and model prompt optimization. (2). *Structured insight mining*. It helps to support 5 open foundation model competitions on data filtering, augmentation, and synthetic data generation with 3,000+ teams [57]. Key lessons learned reveal that standardized and systematic actions provided by `Data-Juicer` (e.g., YAML recipes, probe sampling, visual analytics integration) accelerate data analysis and understanding compared to ad-hoc implementations.

## 6.4 The Case of Medium Scales

**Performance Profile.** When datasets are scaled to 56M samples, processing times increase significantly with the standalone mode, thus, its performance is omitted in Fig. 4e and Fig. 4b. Here, *the Ray mode outperforms in all instances, demonstrating considerable speedups with increasing node counts, making it the recommended choice for medium-scale scenarios*. Moreover, compared to native nodes of Elastic Computing Service (ECS, green lines), the Ray mode on Deep Learning Containers (DLC, red lines) is faster, saving 24.8% of processing time due to the Alibaba Cloud's optimization dedicated to cluster networking.

**Implications & Typical Application Scenarios.** In this scale, `Data-Juicer` boosts data flywheels once we find high-quality data recipes. There have been many synthesis-based and ready-to-use data recipes built upon `Data-Juicer` like [28, 69, 64], where more compute investment brings larger-size

datasets. Several lessons were learned from them: (1) Although the loss functions of foundation models are relatively standardized, we can flexibly inject preferred inductive bias with dedicated data synthesis, such as contrastive learning [28] and data-difficulty based curriculum learning [64]. (2) Foundation models emerge with expert-level knowledge, which can be used as proxy annotators, largely reducing manual labeling costs in benchmark construction [69].

## 6.5   The Case of Large Scales

**Performance Profile.** For datasets at the 70B-sample scale, professional cloud-scale distributed data processing products are advantageous. Users benefit from the vast cloud computing resources without dealing with the intricacies of setup and management. From our experiments, we recommend *Ray-DLC for multimodal recipes and MaxCompute for text-only recipes at this scale*. (1) For multimodal recipes, using 3200 Ray-DLC cores process datasets in 1780.86s and 7083.5s for 500x and 2500x dataset sizes, respectively, indicating good scalability. On the other hand, the MaxCompute engine requires 1.5 times more processing time using the same resources, due to the challenges of loading large-size multimodal data. (2) For text-only recipes (Fig. 4c), although the Ray mode benefits from additional cores even at the scale of ten thousand, MaxCompute is the fastest, requiring about 1/4 of the time and using 1/2 fewer cores. This is attributed to MaxCompute's co-optimization of distributed computing and storage, a feature not as advanced in Ray's implementation.

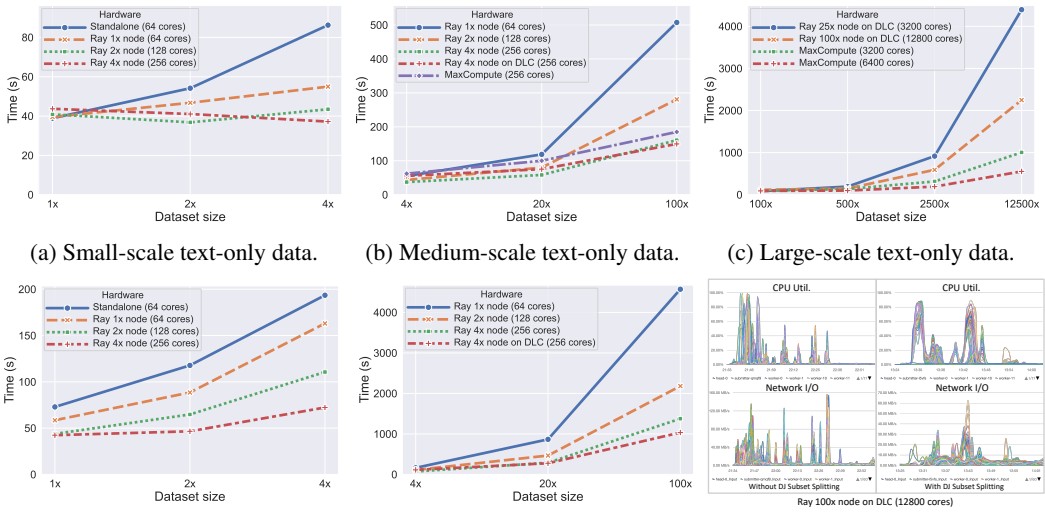

(a) Small-scale text-only data.    (b) Medium-scale text-only data.    (c) Large-scale text-only data.

(d) Small-scale multimodal data.    (e) Medium-scale multimodal data.    (f) Resource utilization comparison.

Figure 4: Processing time comparison across computing engines, dataset sizes, and workloads. A high-resolution version of subfigure (f) can be found in Appendix, Figure. 16

**Impact of I/O and Data Splitting.** I/O is critical for data processing workloads. For distributed processing, large-scale datasets need to be distributed to nodes, stored on shared disks like network attached storage (NAS), or on cloud storage services like Alibaba Cloud object storage service (OSS) and cloud parallel file storage (CPFS). All these introduce inevitable I/O and network communication costs. In our experiments shown in Fig. 4, we use Alibaba Cloud standard CPFS, compared to which NAS and OSS increase time costs by 20-30%. Furthermore, using another AI-oriented CPFS product with a 3x larger network bandwidth and optimized remote direct memory access to process the 12,500x dataset with 3200 cores in Ray mode takes 1,625s, 2.7x faster than the standard CPFS (4,396s). We then scale up the dataset to 125,000x, resulting in times of 14,617s and 7,611s with 3,200 and 6,400 cores, respectively, further demonstrating the efficiency and scalability of the system.

Additionally, the adaptive subset splitting capability for Ray mode, introduced in Sec. 5.1, offers 2x~3x acceleration in our experiments. For example, with 100 Ray-DLC nodes and 12,800 cores, our subset splitting method reduces the processing time of the 12,500x dataset from over 5,000s to about 2,000s. Analysis of Ray's resource utilization shows that its automatic block-splitting strategy incurs high network communication costs and low CPU utilization when many nodes handle a few dataset files. Our strategy pre-splits datasets according to size and node count, optimizing alignment

with Ray's features and enhancing efficiency. As seen in Fig. 4f, this strategy reduces peak network I/O from about 160MB/s to 60MB/s and achieves more consistent CPU utilization across all nodes.

**Large-scale Deduplication.** We tested the MinHash-based `RayDeduplicator` with datasets from Redpajama [18] and Common Crawl chunks [19], sized at 200GB, 1TB, and 5TB, using CPU counts ranging from 640 to 1,280 cores. As shown in Table 2, `RayDeduplicator` efficiently scales with increased data size and computing resources, indicating its capability to handle large-scale deduplication tasks effectively. When the data size in-

Table 2: Deduplication time (minutes) with Ray nodes across varying dataset sizes and CPU counts (160 cores per node).

| # CPU | 200GB Time | 1TB Time | 5TB Time |
|---|---|---|---|
| 4*160 | 11.13 min | 50.83 min | 285.43 min |
| 8*160 | 7.47 min | 30.08 min | 168.10 min |

creases by 5x, the processing time increases by 4.02x∼5.62x. When the number of CPU cores doubles, the processing time decreases to 58.9%∼67.1% of the original time. Notably, `RayDeduplicator` can process 5TB of data in 2.8 hours using 8 Ray nodes (8*160 CPU cores). As a comparison, NVIDIA's NeMo Curator, which leverages cuDF [20] and Dask [52], takes 1.8 hours to process 1.1TB of Redpajama data using 64 A100 GPUs (64*6,192 CUDA cores), as shown in [55].

**Implications & Typical Application Scenarios.** The optimizations in `Data-Juicer` benefit many emerging paradigms to meet the needs of large-scale scenarios. For example, it powers *enterprise foundation model training* from Alibaba Tongyi and Alibaba Cloud's production deployments, especially for TB-token pre-training and costly video/image processing for spatial intelligence. In addition, it helps to explore the data scaling law [25] and reinforced fine-tuning [40], where users need to efficiently process scalable feedback data for learning from experience with environment interactions. Moreover, the key lesson is that complex distributed system introduces non-linear scaling tradeoffs that depend on data modality and access patterns, and storage-compute-software co-design becomes critical beyond 10M samples.

# 7 Conclusions, Limitations & Future Works

`Data-Juicer` 2.0 emerges as a versatile scaffold in the evolution of foundation models, providing efficient and adaptive data processing solutions for handling the diversity and scale of modern datasets. Our re-envisioned architecture leverages multi-layered adaptability to coordinate different modules, operators, and runtime environments. Extensive evaluations reveal its high performance across diverse cloud-scale workloads. By open-sourcing this system, we aim to foster a vibrant community of contributors and users, encouraging collaborative development and driving innovation to underpin the next generation of foundation models.

There are several limitations in `Data-Juicer` 2.0. From the perspective of the computing system, future work includes further enhancing processing adaptability through model-driven agents [24], scalability by optimizing the single-node transmission bottleneck in Ray's head node [56], and efficiency by supporting GPU backend engines like NeMo Curator [27]. Looking ahead, one key direction for future development is enhancing the framework to cover multilingualism, more scenarios (e.g., AI4science, self-driving, embodied intelligence), and a wider range of data governance and safety considerations, such as ensuring data privacy across enterprise-level security protocols. Besides, scaling to the next order of magnitude of processed data is important and requires forward-thinking architectural design like more advanced pipeline optimization, as we discussed in Appendix G.

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

# Appendix

**Table of Contents**

# A List of New Operators in `Data-Juicer` 2.0

The full list of new OPs in `Data-Juicer` 2.0 is shown in Table 3.

Table 3: List of new OPs in `Data-Juicer` 2.0. The full OP list can be found in our document in GitHub repository.

| Modality Type | OP Name (OP Type as the last suffix) | OP Description | Function Type | Implementation Type |
|---|---|---|---|---|
| Image-only | image_blur_mapper | Blur images | Augmentation | Rule-based |
| | image_remove_background_mapper | Remove background of images | Augmentation | Model-based |
| | image_face_blur_mapper | Blur faces detected in images | Privacy | Model-based |
| | image_aesthetics_filter | Keeps samples containing images whose aesthetics scores are within the specified range | Cleaning | Model-based |
| | image_aspect_ratio_filter | Keeps samples containing images with aspect ratios within the specified range | Cleaning | Rule-based |
| | image_face_ratio_filter | Keeps samples containing images with face area ratios within the specified range | Cleaning | Model-based |
| | image_nsfw_filter | Keeps samples containing images with NSFW scores below the threshold | Cleaning | Model-based |
| | image_shape_filter | Keeps samples containing images with widths and heights within the specified range | Cleaning | Rule-based |
| | image_size_filter | Keeps samples containing images whose size in bytes are within the specified range | Cleaning | Rule-based |
| | image_watermark_filter | Keeps samples containing images with predicted watermark probabilities below the threshold | Cleaning | Model-based |
| | image_deduplicator | Deduplicates samples at document-level using exact matching of images between documents | Deduplication | Model-based |
| | ray_image_deduplicator | Deduplicates samples at document-level using exact matching of images between documents on ray | Deduplication | Model-based |
| Image-Text | image_captioning_from_gpt4v_mapper | Generate texts based on GPT-4V for images | Synthesis | Model-based |
| | image_captioning_mapper | Generate texts based on image-to-text models for images | Synthesis | Model-based |
| | image_diffusion_mapper | Generate images based on text-to-image models for texts | Synthesis | Model-based |
| | mllm_mapper | Use multimodal large language models for visual question answering tasks | Wrapper of external tool | Model-based |
| | sdxl_prompt2prompt_mapper | Use the generative model SDXL and image editing technique Prompt-to-Prompt to generate pairs of similar images | Synthesis | Model-based |
| | image_segment_mapper | Perform segment-anything on images and return the bounding box values | Wrapper of external tool | Model-based |
| | sentence_augmentation_mapper | Augment sentences using large language models | Augmentation | Model-based |
| | image_text_matching_filter | Keeps samples with image-text classification matching score within the specified range based on a BLIP model | Cleaning | Model-based |
| | image_text_similarity_filter | Keeps samples with image-text feature cosine similarity within the specified range based on a CLIP model | Cleaning | Model-based |
| | phrase_grounding_recall_filter | Keeps samples whose locating recalls of phrases extracted from text in the images are within a specified range | Cleaning | Model-based |
| | image_pair_similarity_filter | Keeps image pairs whose cosine similarity of features is within a specified range, based on a CLIP model. | Cleaning | Model-based |
| | text_pair_similarity_filter | Keeps text pairs whose cosine similarity of features is within a specified range, based on a CLIP model | Cleaning | Model-based |
| Video-only | video_face_blur_mapper | Blur faces detected in videos | Privacy | Model-based |
| | video_ffmpeg_wrapped_mapper | Wrapper to run a FFmpeg video filter | Wrapper | Tool-based |
| | video_remove_watermark_mapper | Remove watermarks in videos | Cleaning | Model-based |
| | video_resize_aspect_ratio_mapper | Resize video aspect ratio to a specified range | Augmentation | Rule-based |
| | video_resize_resolution_mapper | Map videos to ones with given resolution range | Augmentation | Rule-based |
| | video_split_by_duration_mapper | Split video by duration | Augmentation | Rule-based |
| | video_spit_by_key_frame_mapper | Split video by key frames | Augmentation | Rule-based |
| | video_split_by_scene_mapper | Split videos into scene clips | Augmentation | Model-based |
| | video_human_tracks_extraction_mapper | Extracts human tracks by linking face and body bounding boxes across frames | Synthesis | Model-based |
| | video_human_demographics_mapper | determines demographic attributes (gender, age, race) for face tracks by aggregating frame-level detections. | Synthesis | Model-based |
| | video_human_description_mapper | generates individual-focused videos from body bounding box tracks and processes it for appearance description and simple actions. | Synthesis | Model-based |
| | video_facial_description_mapper | generates face-focused videos from face bounding box tracks and processes them for emotion and facial description. | Synthesis | Model-based |
| | video_active_speaker_detection_mapper | detects active speaking by analyzing face track sequences alongside corresponding audio | Synthesis | Model-based |
| | video_ASR_mapper | generate the automatic speech recognition result for video | Synthesis | Model-based |
| | video_speech_emotion_recognition_mapper | Detect the emotion category of the speech in the video | Synthesis | Model-based |

Table 3: List of new OPs in `Data-Juicer` 2.0. The full OP list can be found in our document in GitHub repository. (Continued)

| | | | | |
|---|---|---|---|---|
| | video_voice_demographics_mapper | analyzes voice demographics (such as gender, age) in video | Synthesis | Model-based |
| | video_aesthetics_filter | Keeps samples whose specified frames have aesthetics scores within the specified range | Cleaning | Model-based |
| | video_aspect_ratio_filter | Keeps samples containing videos with aspect ratios within the specified range | Cleaning | Rule-based |
| | video_duration_filter | Keep data samples whose videos' durations are within a specified range | Cleaning | Rule-based |
| | video_face_ratio_filter | Keep samples whose frame ratio containing faces is greater than a certain threshold are retained. | Cleaning | Model-based |
| | video_motion_score_filter | Keep samples with video motion scores within a specific range | Cleaning | Rule-based |
| | video_nsfw_filter | Keeps samples containing videos with NSFW scores below the threshold | Cleaning | Model-based |
| | video_ocr_area_ratio_filter | Keep data samples whose detected text area ratios for specified frames in the video are within a specified range | Cleaning | Model-based |
| | video_resolution_filter | Keeps samples containing videos with horizontal and vertical resolutions within the specified range | Cleaning | Rule-based |
| | video_watermark_filter | Keeps samples containing videos with predicted watermark probabilities below the threshold | Cleaning | Model-based |
| | video_deduplicator | Deduplicates samples at document-level using exact matching of videos between documents | Deduplication | Model-based |
| | ray_video_deduplicator | Deduplicates samples at document-level using exact matching of videos between documents on ray | Deduplication | Model-based |
| Video-Text | video_captioning_from_audio_mapper | Generate texts for videos according to their audio streams based on audio LLMs | Synthesis | Model-based |
| | video_captioning_from_frames_mapper | Generate texts for videos according to sampled frame images based on image-to-text models | Synthesis | Model-based |
| | video_captioning_from_video_mapper | Generate texts for videos based on video-to-text models | Synthesis | Model-based |
| | video_tagging_from_audio_mapper | Generate tags from audio streams extracted from the video. | Synthesis | Model-based |
| | video_tagging_from_frames_mapper | Generate video tags from frames extracted from the video. | Synthesis | Model-based |
| | video_captioning_from_summarizer_mapper | Generate texts by summarizing several types of generated texts (from video/audio/frames, tags from audio/frames, ...) | Synthesis | Model-based |
| | video_frames_text_similarity_filter | Keep data samples whose similarities between sampled video frame images and text are within a specific range | Cleaning | Model-based |
| | video_tagging_from_frames_filter | Keep samples containing videos with given tags | Cleaning | Model-based |
| Audio-only | audio_ffmpeg_wrapped_mapper | Wrapper to run a FFmpeg audio filter | Wrapper | Tool-based |
| | audio_add_gaussian_noise_mapper | Add gaussian noise to audio. | Augmentation | Tool-based |
| | audio_duration_filter | Keep data samples whose audios' durations are within a specified range | Cleaning | Rule-based |
| | audio_nmf_snr_filter | Keep data samples whose audios' Signal-to-Noise Ratios are within a specified range | Cleaning | Rule-based |
| | audio_size_filter | Keep data samples whose audios' sizes are within a specified range | Cleaning | Rule-based |
| Text-only | calibrate_qa_mapper | Calibrate question-answer pairs based on reference text | Augmentation | Model-based |
| | calibrate_query_mapper | Calibrate query in question-answer pairs based on reference text | Augmentation | Model-based |
| | calibrate_response_mapper | Calibrate response in question-answer pairs based on reference text | Augmentation | Model-based |
| | extract_entity_attribute_mapper | Extract attributes for given entities from the text. | Synthesis | Model-based |
| | extract_entity_relation_mapper | Extract entities and relations in the text for knowledge graph. | Synthesis | Model-based |
| | extract_event_mapper | Extract events and relevant characters in the text. | Synthesis | Model-based |
| | extract_keyword_mapper | Generate keywords for the text. | Synthesis | Model-based |
| | extract_nickname_mapper | Extract nickname relationship in the text. | Synthesis | Model-based |
| | extract_support_text_mapper | Extract support sub text for a summary. | Synthesis | Model-based |
| | extract_tables_from_html_mapper | Extract tables from HTML content. | Cleaning | Rule-based |
| | generate_qa_from_examples_mapper | Generate question and answer pairs based on examples. | Synthesis | Model-based |
| | generate_qa_from_text_mapper | Generate question and answer pairs from text. | Synthesis | Model-based |
| | optimize_qa_mapper | Optimize both the query and response in question-answering samples. | Augmentation | Model-based |
| | optimize_query_mapper | Optimize the query in question-answering samples. | Augmentation | Model-based |
| | optimize_response_mapper | Optimize the response in question-answering samples. | Augmentation | Model-based |
| | pair_preference_mapper | Construct paired preference samples. | Augmentation | Model-based |
| | text_chunk_mapper | Split input text to chunks. | Synthesis | Model-based |
| | naive_grouper | Group all samples to one batched sample. | Synthesis | Rule-based |
| | key_value_grouper | Group samples to batched samples according values in given keys. | Synthesis | Rule-based |

Table 3: List of new OPs in `Data-Juicer` 2.0. The full OP list can be found in our document in GitHub repository. (Continued)

| | | | |
|---|---|---|---|
| entity_attribute_aggregator | Return conclusion of the given entity's attribute from some docs. | Synthesis | Model-based |
| most_relevant_entities_aggregator | Return most relevant entities with the given entity from some docs. | Synthesis | Model-based |
| nested_aggregator | Considering the limitation of input length, nested aggregate contents for each given number of samples. | Synthesis | Model-based |
| llm_quality_score_filter | Keep sample with high quality score estimated by LLM. | Cleaning | Model-based |
| llm_difficulty_score_filter | Keep sample with high difficulty score estimated by LLM. | Cleaning | Model-based |
| domain_diversity_selector | Select samples based on the data's domain diversity. | Cleaning | Model-based |

# B   Showcase of Typical Multimodal Operators

The new OPs include some unique contributions to `Data-Juicer` 2.0, and others partly inspired by SOTA data processing methodologies for foundation models [49, 70, 22]. Below are three representative examples that showcase the diverse computational operations and requirements of these OPs.

`phrase_grounding_recall_filter`: This OP, newly developed by `Data-Juicer` 2.0, assesses alignment and consistency between images and textual descriptions. As illustrated in Fig. 5a, it identifies noun phrases in the text that refer to key entities. Subsequently, an open-vocabulary object detection model, such as Owl-ViT [37], attempts to detect corresponding entities within the image. The OP then calculates the recall of detected phrases to evaluate consistency, where a higher recall indicates better image-text coherence.

`video_motion_score_filter`: This OP quantifies video dynamics. As shown in Fig. 5b, it samples multiple frames at a specified frames-per-second (FPS) rate and computes optical flows. The average magnitude of these flows determines the motion score, with higher scores indicating more dynamic content. To accommodate various computational resources, both a GPU-based RAFT version [60] and a CPU-based OpenCV version [4] are available.

`video_captioning_from_summarizer_mapper`: This OP generates new video captions by combining tagging and captioning capabilities from various OPs. As illustrated in Fig. 5c, it uses six OPs from different angles: two for tagging and captioning the audio stream, two for static visual frame analysis, and two for dynamic video stream and information integration. A summarizer finally incorporates the three captions and two tag sets into a new caption. This composition yields more accurate and comprehensive captions by considering multi-dimensional and multi-perspective content.

# C   User Interfaces

## C.1   Low-level APIs

The framework's core capabilities are exposed through Python-based programmatic interfaces, providing object-oriented logical encapsulation for both the fundamental `Data-Juicer-Dataset`, `Data-Juicer-Operators` and other runtime modules. This design delivers developers maximum flexibility and customizability. Processing workflows can be automatically chained by passing a series of OP instances to a loaded `Data-Juicer-Dataset` object (e.g., `data.process([op1, op2])`), enabling various operations to be performed on the dataset in a single pass. Additionally, the framework supports applying an instantiated `Data-Juicer-Operators` to a target dataset (e.g., `op.run(data)`), enhancing the reusability of OP instances. More details about the `Data-Juicer-Dataset` and `Data-Juicer-Operators` are introduced in Sec. 5 and Appendix D. This dual approach—chained processing and individual OP application—optimizes both efficiency and modularity in data processing tasks, while leveraging the inherent advantages of Python's ecosystem.

## C.2   RESTful APIs

Utilizing standard Python type hints, we provide one-click generation of high-performance web APIs, capable of automatically discovering, registering, and adapting OP classes and related tools. Users can rapidly initiate a web server supporting the Asynchronous Server Gateway Interface by simply executing a service script, eliminating the need for manual code writing. The asynchronous concurrency mechanism enables options such as lightweight background tasks and mitigates potential bottlenecks for endpoints that may experience prolonged network I/O blocking. Each OP is accessible via POST requests, typically executing the OP's `run()` method as the endpoint. The target dataset path is passed through query parameters, with additional configurable OP parameters transmitted via JSON payload. Upon completion, the path to the processed dataset is returned. This invocation through Web APIs allows for a centralized host with distributed access, reducing deployment complexity. It also facilitates the separation of application logic from execution, potentially fostering the development and release of more applications built upon `Data-Juicer`. Importantly, the extensive customization parameters available in the programming API can be seamlessly passed through the Web API,

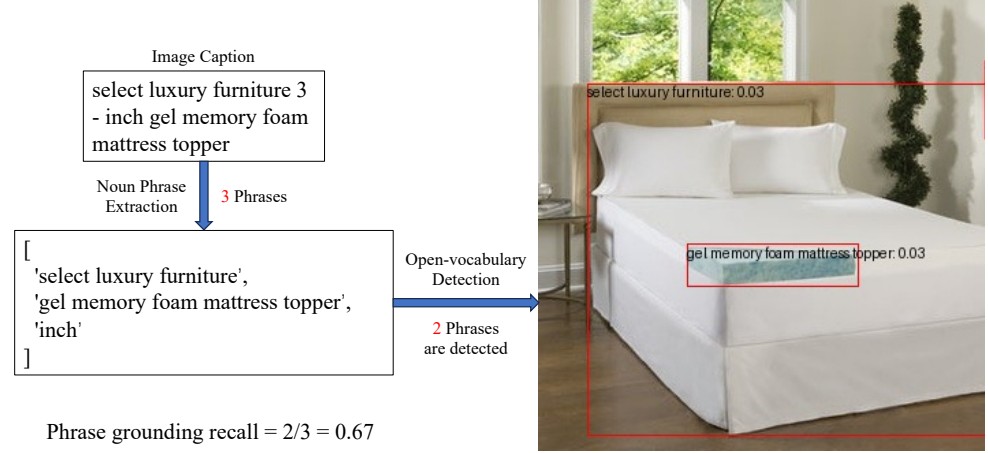

(a) Example of `phrase_grounding_recall_filter`.

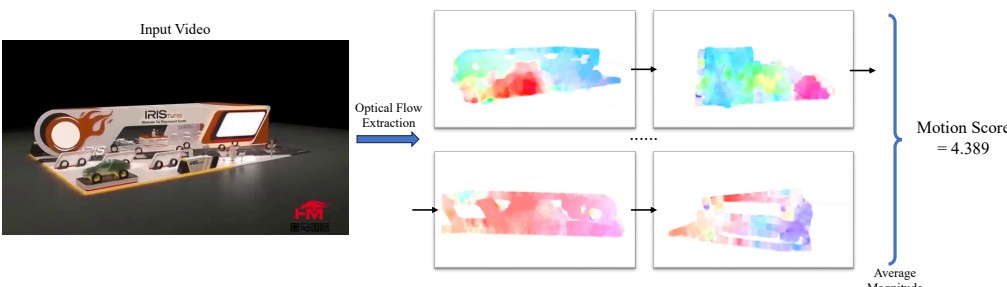

(b) Example of `video_motion_score_filter`.

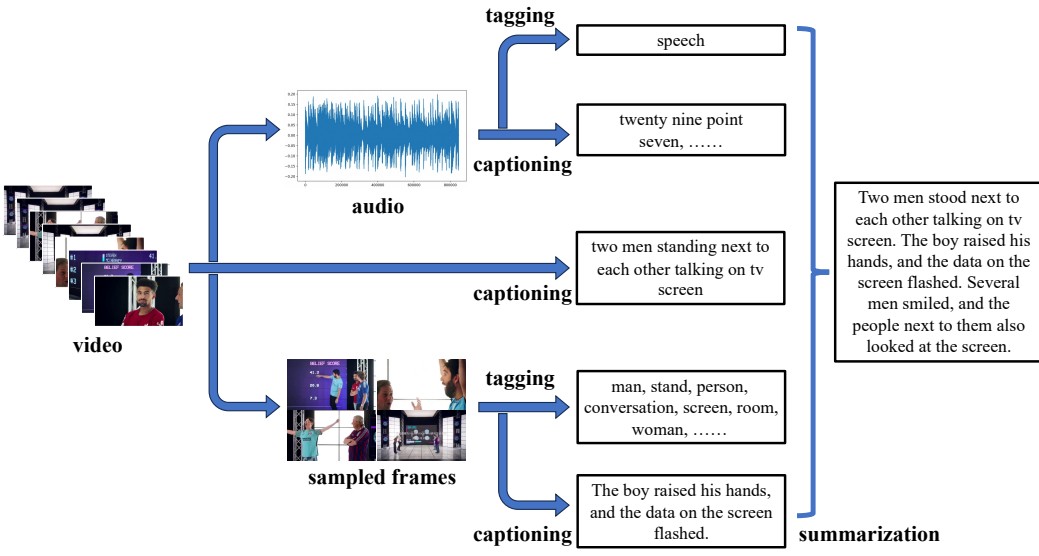

(c) Example of `video_captioning_from_summarizer_mapper`.

Figure 5: Illustrative examples of new operators.

maintaining full functionality without compromising usability, and facilitating serviceful calling by higher-level interfaces as introduced later.

## C.3 Web & CMD tools

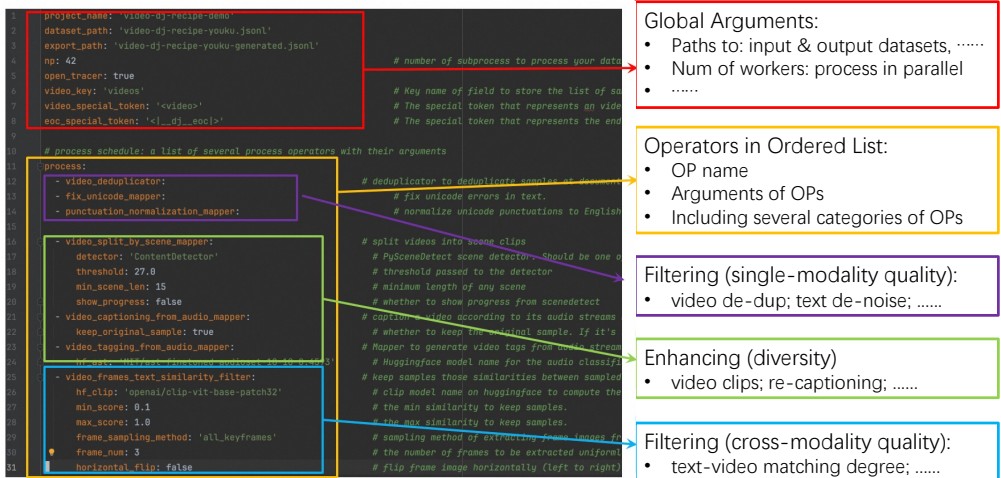

Figure 6: An example of config file for data recipe.

Thanks to the low-level and RESTful APIs, we further establish service-level capabilities across various scenarios in `Data-Juicer` 2.0, encapsulating scenario-specific processing solutions into built-in data recipes, described as YAML configuration files for end-to-end workflows (an example for video-data-synthesis is shown in Fig. 6). Centered around these data recipes, we provide user-friendly, highly encapsulated command-line interfaces as illustrated in Listing 1, supporting automated lightweight installation and recipe-level invocation. These tools automatically gather and install the requisite of OPs within the recipe, perform automated OP fusion, and maximize adaptive utilization of computing resources while minimizing user cognitive load and operational cost.

```
1  dj-install --config recipe.yaml   # install dependencies only for this recipe
2  dj-analyze --auto --dataset_path xx.jsonl  # default OPs, or via specific recipe
3  dj-process --config recipe.yaml   # for both standalone or distributed
```

Listing 1: Representative tools available in `Data-Juicer` 2.0.

Utilizing the aforementioned APIs and tools, `Data-Juicer`'s capabilities are seamlessly integrated into Alibaba Cloud's visual modeling product, PAI-Designer [14]. As shown in Fig. 1, `Data-Juicer-Operators` are encapsulated as visual components within Designer, presented in a directory tree structure for easy access. Users can manage data by intuitive drag-and-drop manners, and import built-in or customized recipes in YAML format using graphical canvas interface. With Designer as entry point, users can activate MaxCompute cluster nodes [13] or containerized deep compute clusters (PAI-DLC) [12] directly for efficient cloud-native distributed data processing, leveraging Alibaba Cloud's data storage solutions (e.g., Object Storage OSS [16], Cloud Parallel File System CPFS [11], and NAS [15]) along with automated performance monitoring and optimization.

## C.4 Natural Language Interaction

```
User: Filter out too short text samples.  The dataset path is input_data.jsonl
Assistant: <thought>Filter out text samples that are too short based on the provided dataset
path.</thought>
<function>run_text_length_filter</function>
<dataset_path>input_data.jsonl</dataset_path>
Assistant: The analyzed statistics are displayed in the image below:  📊
Assistant: Based on the analysis, please enter the minimum and maximum threshold values for filtering.
User: 80 120
System: Execute function run_text_length_filter
[ARGUMENTS] dataset_path:  input_data.jsonl
[STATUS] SUCCESS
[RESULT] Filtered dataset path:  output.jsonl
Assistant: <thought>The filtering has been successfully executed, and the filtered dataset is now at the
specified path.</thought>
<function>finish</function>
<response>Your request to filter out too short text samples has been completed.  The processed dataset
is available at output.jsonl.</response>
Assistant: Your request to filter out too short text samples has been completed.  The processed dataset
is available at output.jsonl.
```

Querying and understanding over 100 diverse OPs have become challenging. Thankfully, advances in foundation models have transformed human-computer interactions. Research shows the synergy of reasoning and acting (ReAct) [65] in language models enhances task-solving. Using RESTful API services, we developed model-driven agents based on ReAct, enabling intuitive data processing through natural language. We adopted AgentScope [24], a multi-agent platform, for low-code integration, using prompt-optimized ReAct agents to align OP functionalities with our RESTful APIs. Built-in function preprocessing, response parsing, and fault-tolerance streamline development. `Data-Juicer` 2.0 thus allows flexible instruction input, interpreting user intents to execute appropriate data processing functions.

An example is shown above for interactive text length filtering using this agent, where users describe tasks with simple and vague language. Specifically, the user requests "filter out too short text samples from $input\_data.jsonl$". The agent's ReAct prompt helps analyze and reason user intent, signaled by the $<thought>$ tag. The $<function>$ tag notes the chosen function, while $<dataset\_path>$ represents parameter mapping. Initially, a statistical analysis of text lengths is visualized, and the agent asks for filtering thresholds. After user input, filtering runs with parameters via RESTful API. Upon completion, the agent confirms the processed dataset path as $output.jsonl$, marks the task finished, and informs the user. Gray dialogs reveal the agent's reasoning and actions, providing insights into the workflow. This example illustrates step-by-step user understanding, function selection, parameter population, and task execution, emphasizing transparency and automation of `Data-Juicer` 2.0 with agentic processing.

## D  Core Runtime Implementation Details

### D.1  Execution Mode Configuration

The core runtime of `Data-Juicer` 2.0 is implemented in Python, featuring a flexible data processing pipeline characterized by two execution modes: a standalone mode for convenient single-node execution, based on the Hugging Face Dataset [32], and a distributed mode that offers scalability across multiple nodes, leveraging the Ray Data [38] and MaxFrame-DataFrame [13]. These frameworks offer diverse computational engines, each with distinct capabilities and suitable use cases, but they also come with heterogeneous programming interfaces and complex implementation details.

In `Data-Juicer` 2.0, we exploit the strengths of these diverse computational engines and their associated programming classes by abstracting a top-level `Data-Juicer-Dataset` class and devising a standardized data representation schema (detailed in the next subsection). The primary design principle employed is the *Facade Pattern*. The abstracted class provides comprehensive and unified development interfaces, facilitating seamless use across standalone, distributed, and cloud cluster environments, meanwhile optimized to support extensive multimodal data processing.

```
1  # Differernt input sources, such as from local files, dataset hubs from Hugging Face and Modelscope, and
   ↪  special types like arXiv, wiki, CommonCrawl, etc.
2  data = DJDataset.load(src_file)
3
4  data = data.process(op1)  # run a single operator
5  data = data.process(op1).process(op2)  # run multiple operators continuously
6  data = data.process([op1, op2])  # run a list of operators
7
8  data.export(tgt_file)  # tgt_file can be either local path or remote path
```

Listing 2: Some basic interfaces of `Data-Juicer-Dataset`.

Moreover, following the principle of *Template Method Pattern*, the `Data-Juicer-Dataset` class establishes templated workflows for OP executing on datasets and provides several basic and unified interfaces, catering to users with diverse programming preferences, as demonstrated in Listing 2. The complexities of underlying computational engines and cumbersome runtime issues related to instantiated dataset objects are abstracted away from these interfaces, ensuring transparency and ease of use for end-users and developers alike.

New functionalities can be added by merely implementing new OP classes without altering the internal logic of `Data-Juicer-Dataset`. The standalone and distributed execution modes automatically recognize and switch configurations based on runtime settings. Coupled with unified *function signatures*, the data processing can be consolidated into a batched processing mode, not only standardizing the *data interface* for OP processing but also facilitating the implementation of a robust sample-level fault tolerance mechanism. The internal adaptation features are elaborated in Appendix E.2.

## D.2 Data Schema Implementation

In the context of foundation models and post-tuning tasks, datasets are complex, originating from various sources and modalities. Target tasks are diverse, requiring different data formats and organizational structures. Integrating diverse data processing into a unified pipeline is challenging. `Data-Juicer` 2.0 adopts an extensible and intermediate data representation schema that intrinsically supports multimodal data types and flexible data field customization.

An example of a data sample is illustrated in Listing 3. In the data schema, each dataset encapsulates column-stored samples represented by a non-recursive dictionary whose level-1 fields include three categories: (1) the core contents within customizable fields, including "text" that is usually used in the pretraining task, and "query", "response", "history" fields represent dialogs in post-tuning tasks, which are directly related to the pretraining or post-tuning procedures in the downstream usage of the dataset; (2) the extra data contents contain the path lists of multimodal data for multimodal datasets; (3) the "meta" information concerning this sample that stems from either its raw data or `Data-Juicer`'s Mappers for tagging, and the "stats" information calculated by `Data-Juicer`'s Filters (all Filter OPs consist of a `compute_stats()` method, utilized for subsequent analysis and conditional filtering).

For multimodal datasets, the content is primarily centered on the textual modality within the "text" field, capable of optionally including several text chunks split by customizable end-of-chunk tokens, which are `<|__dj__eoc|>` in default. Each chunk serves as a semantic unit, where the associated multimodal data within the same chunk relates to the same topic, thereby aligning with one another. Multimodal data (excluding text) are denoted by ordered customized special tokens in the text (e.g., `<__dj__image>`), storing entities at designated files, accessible through indexed file paths within the modality fields of samples to facilitate object sharing and elimination of redundant computing. For instance, in the example shown in Listing 3, the first chunk in the "text" field split by the `<__dj__eoc>` token contains the first `<__dj__image>` token and a description to this image, which corresponds to the first image stored in the "images" field.

This token-centric representation elegantly aligns multimodal data while preserving positional information, friendly to current prevalent learning paradigms for foundation models employing next-token

```
1  {
2    // >>> core contents: texts, dialogs, ...
3    // for general pretraining
4    "text": "<__dj__image> desc of image 1 <|__dj__eoc|> desc of image 2 <__dj__image> <|__dj__eoc|>",
5    // for post-tuning
6    "query": "<query2>",
7    "response": "<response2>",
8    "history": [["<query1>", "<response1>"]],
9    // <<< core contents
10
11   // >>> extra data contents: multimodal data paths, ...
12   "images": [
13     "path/to/the/image1",
14     "path/to/the/image2"
15   ],
16   // <<< extra data contents
17
18   // >>> meta infos and stats, which could be primitive or produced by Data-Juicer
19   "meta": {
20     "src": "customized",
21     "version": "0.1"
22   },
23   "stats": {
24     "lang": "en",
25     "image_widths": [224, 336]
26   },
27   // <<< meta infos and stats
28 }
```

Listing 3: Illustration of data schema of `Data-Juicer` 2.0.

prediction tasks [3, 62]. And it's compatible with both simple cross-modal pairing datasets (e.g., image-text pairs [10] and video-text pairs [63]), as well as complex interleaved multimodal datasets (e.g., MMC4 [70]), owing to its chunkable and special token design.

For post-tuning datasets, we provide several core fields in the core contents to represent the dialog datasets in our intermediate format, which is naturally aligned with Alpaca and Query-Response formats in the ModelScope-SWIFT repository [67] for smooth training of 400+ foundation models. As for other widely-used formats in well-known repositories like LLaMA-Factory [68], a suite of dataset conversion tools was developed to efficiently transform various popular multimodal datasets in them to and from the `Data-Juicer` schema, enabling users to process extensive existing models and datasets within `Data-Juicer`. These tools also serve as demonstrations for extending support to other uncovered datasets.

### D.3 Operator Factory & Taxonomy

#### D.3.1 OP Design Principles

Operators are fundamental units responsible for executing processing functionalities, such as enhancing the accuracy of descriptive text or filtering out images containing Not-Safe-for-Work (NSFW) content. As illustrated by the middle orange box in Fig. 1, `Data-Juicer` defines five previous atomic OP classes: Formatter, Filter, Mapper, Deduplicator, and Selector; alongside four new types of compositional OPs: Grouper, Aggregator, FusedOP, and ScriptOP. The first five OP classes handle dataset format conversion, sample filtering, modification, deduplication, and selection, respectively. Following the *Strategy Pattern*, they are designed to encapsulate diverse algorithms that can be dynamically used to process the data. Each OP has a clearly defined role and can be interchanged or modified without impacting other system parts.

Moreover, following the *Decorator Pattern*, compositional OPs are provided to enhance existing functionality without modifying the prevailing OP structure while dynamically adding data processing behavior to objects. The FusedOP enables explicitly grouping multiple atomic OPs to process data

```
1  # Default approach processes the whole dataset sequentially
2  data.process(op1).process(op2) # or data.process([op1, op2])
3
4  # Explict fine-grained processing at the batch level
5  data.process(FusedOP([op1, op2], bs=1000))
6  # Internally
7  for data_batch in data.next(batch_size):
8      data_batch.process(op1).process(op2)
```

Listing 4: Illustration of FusedOP in `Data-Juicer` 2.0.

in a fine-grained manner within the same data batch, opposed to the default sequential processing across datasets as demonstrated by Listing 4. The Grouper takes a `Data-Juicer-Dataset` as input and groups data samples into batches, which can then be input into the Aggregator for subsequent aggregation. For example, we can employ *extract_entity_attribute_mapper* and *entity_-attribute_aggregator* following a *key_value_grouper* for meta-information extraction from textual input. Meanwhile, the ScriptOP allows users to incorporate existing Python files or execute code snippets, utilizing customized functionalities encoded within scripts like `helper_func.py`. This includes leveraging existing `Data-Juicer` command tools or integrating short Python scripts (e.g., lambda functions).

Together, these nine OP types provide robust expressive capabilities for end-to-end data processing solutions that can be embedded within a single YAML configuration file (as depicted in Fig. 6).

### D.3.2 Unified Coordination of Logical Operations

In `Data-Juicer` 1.0, the logical operations of different OPs were coordinated within various executors (either standalone or distributed) rather than being bound to the OPs themselves. Disentangled from the executor's scheduling interface, it becomes challenging to determine the execution logic of different OPs, making the development and extension of individual OPs less intuitive and lacking self-explanatory qualities. In `Data-Juicer` 2.0, several design principles are utilized to address this issue, including the *Abstract Factory Pattern, Template Method Pattern, and Single Responsibility Principle*.

Specifically, a top-level OP factory class is abstracted above the aforementioned fundamental OP classes. In this class, functionalities common to all OPs are extracted, such as preprocessing of instantiated parameters, support for serialization, and configuration of OP-aware runtime parallelism. Besides, a unified `run()` method is implemented, maintaining a consistent interface for integration and API calls. Furthermore, parallelism in multi-processing and multi-GPU is automatically configured and decoupled from specific OPs, ensuring transparency for end users and developers, as introduced subsequently in Appendix F. Lower-level OP classes define their own templated execution logic behind the run invocation. Taking the Filter class as an example, its core logic first engages the `compute_stats()` method to obtain statistical information based on specific metrics and then invokes its `process()` method to determine sample filtering based on thresholds.

On the one hand, this simplifies users' understanding of OP types. Users can instantiate any OP and invoke it with a unified parameter signature using `op.run()`, thereby reducing the learning curve. The templated execution flow of various OP types is self-contained within base classes, eliminating dependencies on external executors to oversee invocation logic. On the other hand, by templating execution logic within base classes, developers can readily modify, extend, or implement new OPs. For instance, a developer aiming to customize an existing Filter does not need to rewrite a new class entirely but can inherit from a related existing class and override specific methods such as `compute_stats()` or others as required.

Regarding the naming and implementation of specific leaf OP classes, we adhere to extracting functionalities that are not tightly coupled with the OPs into common utility classes wherever feasible. This approach enables each OP to focus on its specific modalities and functionalities, facilitating a reduction in code complexity and enhancing clarity in understanding individual OP classes. Compared to previous implementations, the revised OP classes demonstrate easier inspection, integration, and

testing. Users can utilize these robust OPs and seamlessly integrate them into their own tools or systems flexibly, both in source code or exposed RESTful API.

### D.4  Control Panel Implementation

With the fundamental `Data-Juicer-Dataset` and `Data-Juicer-Operators` primitives established, we offer a series of control panel modules (the red box in Fig. 1) to organically combine them and accomplish data processing tasks. `Executor` encapsulates a series of standardized execution processes tailored for different standalone and distributed engines, leveraging modules such as `Config` and `Monitor` to accomplish end-to-end system configuration, data loading, analysis, processing iteration, data checkpointing, and more. Template workflows manage the complete data development lifecycle, including feedback, data storage, logging, and performance and operational monitoring.

To avoid the substantial costs from trial and error for data and model development in foundation model scenarios, we further develop a Sandbox suite in `Data-Juicer 2.0` for *data-model co-development*, serving as a specialized intermediate layer connecting data processing jobs to numerous open-source infrastructures of model training and evaluation. The suite offers template workflows that extend beyond dataset-only development by incorporating cost-effective model training and evaluation signals, and quantitatively studying interplays among data, model, and compute. Users can easily conduct small, quick, and comparative experiments to find insights and superior data recipes, which can then be scaled to larger models and datasets, thereby optimizing the return on investment in data-model co-development. Additional details and empirically validated support can be found in [9].

## E  Optimization Details on `Data-Juicer`-**Dataset**

### E.1  Engine-agnostic Processing

In `Data-Juicer 1.0`, the data processing pipeline is implemented using two distinctive execution modes: a standalone mode tailored for single-node operations and a distributed mode designed for scalability over multiple nodes. These modes exploit different dataset classes, each with a unique set of functionalities and interfaces. The default standalone execution mode employs the Hugging Face `Dataset` class [32], which provides a rich array of encapsulated functions, such as `dataset.map()` and `dataset.filter()`. It's also equipped with configurable batch processing capabilities essential for various computational needs. In contrast, the distributed execution mode leverages the `Ray Dataset` class [38], which scales effectively across multiple nodes.

Despite both dataset classes using a storage format based on Apache Arrow [1], they exhibit significant differences in the behaviors and interfaces exposed. For instance, the Ray Dataset delineates individual and batched sample processing using separate methods: `map` and `map_batches`. It also allows to specify GPU counts for optimized scheduling. Meanwhile, the Hugging Face Dataset excels in supporting a broad spectrum of data modalities, such as image and audio. Moreover, the Hugging Face Dataset is typically applied in read-heavy data processing scenarios, such as in-memory tokenization and tensor reshaping, which are crucial for training deep learning models. For processing tasks in the context of foundation models, especially those involving synthesis operations, write-heavy procedures warrant attention. Here, the Ray Dataset provides flexible data exportation techniques advantageous to such tasks.

As highlighted in Sec. 5.1, we introduce a top-level `Data-Juicer-Dataset` class in `Data-Juicer` 2.0, along with common functions to bridge the variety of interfaces and implementation specifics across diverse computational engines. Besides the support of Hugging Face Dataset and Ray Dataset, our `Data-Juicer-Dataset` is also seamlessly extendable to support distributed computing within Alibaba Cloud ecosystems, thanks to the compatibility between ours and MaxFrame-DataFrame classes with intermediate in-memory formats like Pandas [41] and NumPy [26] or external-memory formats like Parquet. Thus, we develop unified wrappers that extract the core execution functions of different `Data-Juicer-Operators` into User-Defined Functions for the MaxCompute engine. Compared to Spark, MaxCompute is compatible in both syntax and runtime on Alibaba Cloud nodes and can be considered a commercially optimized version. An internal empirical comparison shows that MaxCompute SQL achieves 50% better performance than native Spark SQL.

**The Deduplicators within** `Data-Juicer` **2.0.** Fuzzy deduplication is complex, involving a mixture of operations such as map, filter, group by, aggregate, and join [6]. The support and performance of

these operations vary across different engines, especially in large-scale scenarios. To demonstrate the engine-agnostic feature of `Data-Juicer` 2.0, we use our *minhash_deduplicator* as an example, which supports the aforementioned three different engines. Users only need to specify algorithm-specific parameters such as `jaccard_threshold` and `num_permutations`. These parameters remain consistent regardless of the engine used, while engine-specific details and optimizations are completely transparent to the user. For instance, we utilize Ray Actors to implement our Ray-based deduplicator, starting with a load-balanced, distributed union-find algorithm [30] and introducing a hash-based aggregation process to enhance memory utilization and efficiency. This method avoids fragmented unions caused by Ray's native `groupby` operation, which is computationally expensive in typical LSH implementations with traditional big-data engines [39]. As a result, we achieve a 3.3x speedup over our vanilla Ray version.

## E.2 Fault Tolerance

In practical processing scenarios, datasets often contain schema-incompatible or corrupted data elements, such as improperly formatted JSON objects or damaged images that cannot be read. This issue becomes increasingly important with large-scale datasets, as processing tasks may extend over several hours or even days. In `Data-Juicer` 1.0, corruption of a single sample would halt the entire processing task, resulting in a waste of computational resources and the loss of already processed samples.

To address this issue, `Data-Juicer` 2.0 introduces a sample-level fault tolerance mechanism designed to enhance processing reliability by providing a worst-case guarantee. A unified exception manager is implemented to automatically capture runtime errors with customizable handlers during the processing of each OP. By default, dataset processing operations are performed at the data-batch level for a general handler. As a result, the system can easily bypass problematic samples by skipping the affected batch (as demonstrated in Fig. 7), while actively tracking and reporting these cases for subsequent debugging and correction. This ensures a seamless user experience and minimizes retry costs in scenarios involving large-scale data processing. Users have the flexibility to either discard these samples or mark them for future reprocessing.

Figure 7: Illustration of fault tolerance for "bad" data.

Of particular note is the fact that implementing this mechanism in `Data-Juicer` 1.0 presented challenges due to the rigid schema consistency mandates imposed by the underlying Hugging Face and Ray dataset classes. The newly refactored batching interfaces facilitate the unified construction of compatible empty samples based on schemas accessed during exception handling, thereby enabling seamless integration with valid dataset entries.

## E.3 Streaming I/O and Subset Splitting

Memory constraints frequently emerge as a bottleneck in data processing tasks associated with foundation models. Memory demands can be substantial, and its precise usage is difficult to predict beforehand. Influential factors include the actual storage demands of objects such as text, image, and audio within individual dataset samples, as well as the storage demands of auxiliary or newly generated objects at runtime. These can stem from varying OP specifications, model sizes, synthesis data volumes, intermediate variables, the number of concurrent processes, and specific computational engines.

To effectively adapt to a diverse range of data processing scenarios with varying data volumes and available computational resources, we introduce streaming loading and data pre-splitting capabilities in `Data-Juicer` 2.0. They collectively facilitate improved computational resource utilization, and provide potential for a flexible programming space of hybrid stream and batch processing.

Firstly, `Data-Juicer` 2.0 offers a streaming loading interface, addressing the current lack of native support in the Arrow framework underlying Hugging Face and Ray Datasets for streaming JSONL data. As many foundation model datasets use JSONL, we developed an in-house patch to alleviate Out-of-Memory (OOM) issues.

Secondly, we develop a user-friendly script that automatically pre-splits the original dataset based on two observations: (1) The size limit of the underlying Apache-Arrow block, and (2) The inherent automatic block-splitting strategy in Ray. When there are tens of thousands of nodes but with only a few dataset files, Ray would split the dataset files according to the available resources and distribute the blocks of the dataset to all nodes, which brings a huge network communication cost and decreases the CPU utilization of each node. Thus, we split the original dataset into smaller 128MB files in advance automatically according to the dataset size and the number of distributed nodes, trying to adapt the features of Arrow and Ray for better performance. This approach reduces location and reprocessing costs associated with fault tolerance and helps mitigate network exchange overheads, especially beneficial in contexts involving large-scale multimodal data, as well as in scenarios that require handling global objects of Ray Actor in distributed modes.

## F   Implementation of Adaptation for `Data-Juicer`-Operator

This section explores the internal adaptations developed for `Data-Juicer-Operator`, which are crucial for optimizing resource allocation and user experience without requiring users to understand hardware specifics or implementation details. We implement several automatic adaptation features for resource management, aiming at balancing resource constraints and operational efficiency within `Data-Juicer` 2.0. These strategies include automatic OP reordering at the recipe level (Appendix F.1), automatic parallelism at the OP level (Appendix F.2), and automatic data insight mining to assess the impact of each OP on data samples, considering both upstream and downstream OPs (Appendix F.3). These features are encapsulated in a dedicated `Adapter` class, which uses a `probe_small_-batch()` method to systematically probe and analyze essential information by applying individual OPs on randomly sampled data in runtime, with default sample size as `min(1000, remaining_-data_size)`.

### F.1   Workloads-aware OP Reordering

At the recipe level, we introduce a new probe-based OP reordering strategy. In `Data-Juicer` 1.0, an OP fusion optimization was proposed to eliminate redundant computations for the Filter OPs, which involved three key steps: detection and grouping of fusible OPs, OP fusion, and OP reordering. The reordering strategy aimed to position more time-consuming OPs at the end of the group to process fewer samples to save time, as some were filtered out by preceding OPs. It is assumed that the fused OPs are the most time-consuming, and only the fused OPs are moved to the end of each group.

However, the assumption is not always correct and the prior reordering strategy omit the unfused OPs, rendering it greedy and suboptimal, especially when applied to diverse datasets and data recipes characterized by varied data distribution and OP orchestration. In `Data-Juicer` 2.0, we advance the reordering strategy to an adaptive approach, which is workload-aware and can be applied automatically to unfused OPs as well.

Specifically, before processing the full dataset, `Data-Juicer` 2.0 utilizes the probe functionality of `Adapter` to obtain estimated processing speeds for individual OPs relevant to specific input datasets. When processing the entire dataset begins, OPs in each group (including the unfused ones) are reordered based on the probed speeds and the commutativity of the Filter OPs. For the fused OP, assuming that there are $n$ fusible OPs in it and their probed speeds are $v_i$ where $i \in \{1, 2, \cdots, n\}$, the estimated speed of the fused OP can be calculated as:

$$v_{fused} = \frac{N}{T_{total}} = \frac{N}{\sum_{i=1}^{n} \frac{N}{v_i}} = \frac{1}{\sum_{i=1}^{n} \frac{1}{v_i}},\tag{1}$$

where $N$ is the total number of samples to be processed by the fused OP and $T_{total}$ is the total time cost of it. Then, faster OPs are prioritized, while slower ones are deferred to later stages. This probe-based approach identifies the globally optimal reordering solution for each OP group, outperforming the suboptimal strategies of the previous version, as empirically validated in Appendix H.2.1.

### F.2  Automatic Operator-wise Parallelism

In data processing for foundation models, it is crucial to recognize that different OPs require vastly different computational resources. Model-based OPs often need several gigabytes of GPU memory, while simple rule-based OPs may only need minimal CPU processing. Therefore, using a uniform parallelism granularity across all OPs in a data pipeline can cause OOM issues for some and resource underutilization for others. To address this challenge, we introduce several automatic mechanisms for OP-wise parallelism.

- Model-based OPs that integrate substantial models require considerable computational time, potentially spanning hundreds of hours on CPUs for large datasets. To mitigate this, `Data-Juicer` 2.0 expedites these OPs by automatically leveraging CUDA and GPUs when available. Given the diverse memory requirements of large models, we utilize the `Adapter` component to conduct quick VRAM benchmarking prior to full-size dataset processing. This information is systematically assigned to the `mem_required` parameter of the respective OPs. During the execution of these OPs on datasets, `Data-Juicer` 2.0 continuously assesses the available GPU memory of the execution environment to dynamically and adaptively determine the optimal parallelism strategy. To further alleviate GPU memory demand, we also integrate quantization libraries such as vLLM [31] to enable efficient inference of leveraged models.

- For non-model-based OPs, attributes such as `cpu_required` and `mem_required` are crucial. In `Data-Juicer` 2.0, users can specify a global parallelism level, typically aligning with the available processor count. Meanwhile, we calculate the adaptive number of processors to finely optimize the entire processing pipeline, aiming to maximize resource utilization (90% by default) as much as possible. For this purpose, the `Adapter` is also employed dynamically and instrumental in determining precise `cpu_required` and `mem_required` values at runtime.

- For general-purpose and I/O-intensive OPs, `Data-Juicer` 2.0 enables batched processing, using a robust default `batch_size` parameter guided by performance saturation trends shown in Appendix H.2.3. Batched processing reduces I/O overhead, boosts efficiency per processor, and enhances overall speed. Additionally, we introduce hierarchical parallelism for OPs involving multimodal data I/O, such as $image\_aspect\_ratio\_filter$. These OPs utilize multi-process and GPU parallelism, along with multiple threads, to handle data batches more efficiently, taking the concurrent opportunities between I/O and computation latencies.

### F.3  Insight Mining Example

Here we provide an example of OP-wise insight mining in Fig. 8.

## G  Future Directions in Pipeline Optimization for Exascale Data

As data processing demands for foundation models scale towards the exabyte level, merely adding more computational resources becomes economically and technically unsustainable. Our future work thus focuses on a next-generation pipeline optimization framework that functions akin to a query optimizer for deep learning data pipelines. This framework is designed to intelligently rewrite and execute data-processing graphs to minimize fundamental bottlenecks in I/O, memory usage, and computation. Key architectural directions include:

**Advanced Operator Fusion.** Building upon the existing filter fusion, this strategy involves analyzing the pipeline's Abstract Syntax Tree (PipeAST) to identify and merge consecutive, computationally compatible operators. For example, a sequence of resize, crop, and normalize operations on an image could be fused into a single, highly optimized kernel. This significantly reduces intermediate data materialization and memory traffic, streamlining per-sample transformations.

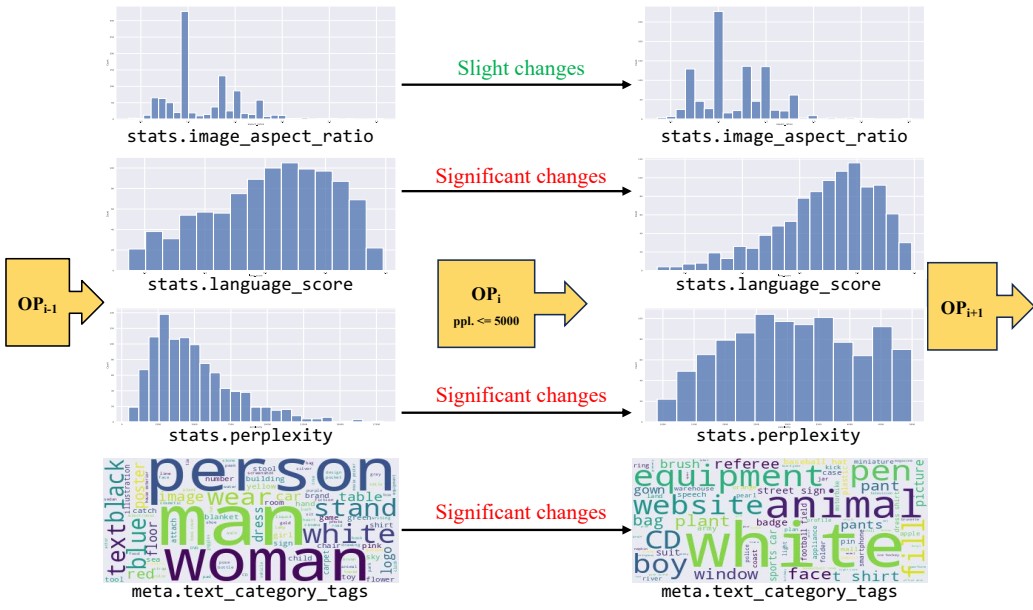

Figure 8: Illustration of OP-wise insight mining.

**Pipeline-Aware Column Pruning.** For wide-schema datasets, such as Parquet files from autonomous driving logs containing hundreds of sensor columns, loading all data is highly inefficient. We envision a system that performs static analysis on the entire pipeline's data dependencies to determine the precise subset of columns required for the complete workflow. This enables selective loading of only the necessary data from storage, drastically reducing I/O and memory footprint from the very first read.

**Predicate Pushdown to the Storage Layer.** This powerful optimization involves pushing inexpensive filtering predicates as close to the data source as possible. For instance, a filter on image metadata (e.g., resolution > 1024x1024 or creation_date > 2023) can be applied during the initial file listing or manifest scanning phase. This "pre-rejects" samples before their potentially large data payloads (e.g., multi-megabyte images) are downloaded or deserialized, preventing immense wastage of network bandwidth and compute resources.

**Dynamic and Adaptive Resource Allocation.** Transcending static resource configurations, this direction focuses on a runtime that monitors real-time system metrics (CPU utilization, GPU VRAM, memory bandwidth). By leveraging advanced features in modern execution engines (e.g., Ray's compiled execution graphs), the system can dynamically adjust operator-level parallelism and resource assignments. For example, it could temporarily allocate more GPUs to a bottleneck model-based filter, ensuring sustained optimal throughput across heterogeneous hardware and fluctuating workloads.

# H  Additional Experiment Details and Results

## H.1  Implementation Details of Scalability Experiments

We start with a base dataset comprising 560k image-text pairs (about totally 5.6 million textual tokens) from the pertaining dataset of LLaVA [36]. We expand this dataset by factors of {2, 4, 20, 100, 500, 2500, 12500, 125000}, resulting in nine datasets scaling to 70B data samples. These datasets are categorized into three scales: small (1x, 2x, 4x), medium (4x, 20x, 100x), and large (100x, 500x, 2500x, 12500x, 125000x). For different scales, we prepare both multimodal and text-only data processing recipes, each containing 5 OPs. We run the recipes on these datasets using various computing engines (Standalone, Ray, MaxCompute) with different Alibaba Cloud resources, including ECS instances, PAI-DLC, and MaxCompute, spanning 1 to 100 nodes and 64 to 12,800 CPU cores. To ensure a fair comparison, we use the same worker node configuration, such as CPU frequency, across different computing engines.

## H.2   Ablation Study on Runtime Adaptations

### H.2.1   Automatic workloads-aware OP reordering

We evaluate the performance improvements due to our probe-based OP reordering on two recipes with different numbers of OPs. The simple recipe contains 5 OPs, with 2 fusible OPs, while the complex recipe includes 13 OPs, with 5 fusible OPs. We run these recipes on the base dataset used before (560k image-text pairs), comparing processing times for all OPs, fusible OPs, and other OPs.

Fig. 9 illustrates the results. Generally, OP fusion is effective for accelerating data processing, and automatic OP reordering offers further improvements based on OP fusion. Notably, OP reordering is more effective in complex data recipes (the right sub-figure), saving more time, especially for fusible OPs (46.09% v.s. 70.22%).

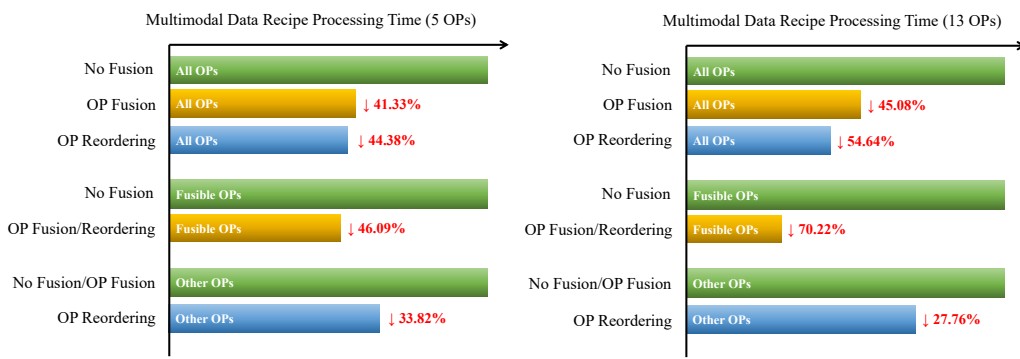

Figure 9: Processing time for simple and complex recipes.

### H.2.2   Automatic GPU resource allocation

Efficient model inference on GPUs is critical for the processing speeds of model-based OPs. We select four image-related OPs, including an image-only OP and three image-text cross-modality OPs, integrating various models that require different GPU memories. Using a test dataset of 10k image-text pairs, we process these OPs on a machine with 64 processors and 8 Nvidia A100 GPUs, and compare the times with and without GPU support in `Data-Juicer` 2.0. Table 4 shows these comparative results.

Table 4: Multimodal OPs processing time comparison between CPU cores and GPUs on 10k samples. "np" denotes the number of processors.

| Multimodal OPs | VRAM | np | Processing Time | |
|---|---|---|---|---|
| | | | CPU | GPU |
| *image_captioning_mapper* | ~16GB | 32 | 8668.87s | 305.44 s |
| *image_diffusion_mapper* | ~8GB | 64 | 100 h | ~1 h |
| *image_text_similarity_filter* | ~1.5GB | 64 | 73.04 s | 35.84 s |
| *image_nsfw_filter* | ~1GB | 64 | 102.59 s | 39.74 s |

As the results show, using GPUs saves at least 50% of processing time for all selected model-based OPs, especially for large, slow models like BLIP-2 [33] used in *image_captioning_mapper* and SDXL model [48] in *image_diffusion_mapper*. Compared to the CPU version, GPU usage and adaptive resource allocation are much more efficient and necessary. Due to the GPU memory limit of around 80GB, we can only allocate at most 4 models of *image_captioning_mapper* OP on a single GPU, so the number of processors for this OP is automatically reduced to 32, preventing OOM errors.

### H.2.3 Batched data processing

We compare the efficiency of single-sample processing in the previous `Data-Juicer` 1.0 and batched processing in `Data-Juicer` 2.0. We use the previous 560k multimodal dataset, processed by 16 processors. Each processor handles about 30k samples. We select four recipes representing different scenarios: Filter-Heavy, Mapper-Heavy, Text-Heavy, and Multimodal-Heavy. Each recipe contains four OPs of the specified "heavy" type and one of another type. We run these recipes with five different batch sizes and summarize the results in Fig. 10a. From these experiments, we conclude: (1) Batched processing is always more efficient. Larger batch sizes consistently speed up data processing in all scenarios, reducing processing time by 84%. (2) Efficiency gains from larger batch sizes plateau beyond a certain threshold. Specifically, batch sizes of 100 or more yield similar benefits. (3) A batch size of 1000 is recommended, generally showing the most efficient processing in our experiments. Consequently, 1000 is selected as the default batch size in `Data-Juicer` 2.0.

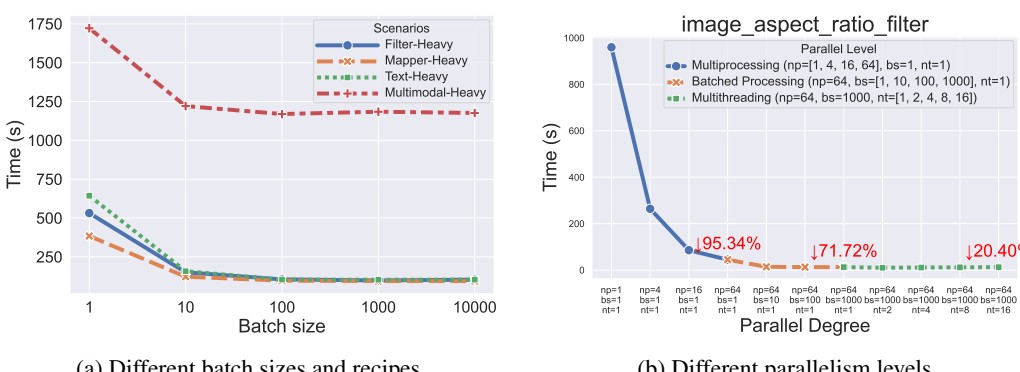

(a) Different batch sizes and recipes.  (b) Different parallelism levels.

Figure 10: Data processing efficiency of OP-wise hierarchical parallelism. "bs" denotes the batch size, and "nt" dennotes the number of threads.

### H.2.4 Automatic OP-wise hierarchical parallelism

To verify the efficiency of the automatic OP-wise hierarchical parallel strategy, we select the multi-modal OP *image_aspect_ratio_filter* as an example. We conduct experiments with various processor counts, batch sizes, and thread numbers, covering three parallel levels: multiprocessing, batched processing, and multithreading. This OP processes approximately 560k image-text pairs. Fig. 10b demonstrates the time consumption and efficiency gains.

As depicted, significant speed improvements are consistently achieved across all parallel levels. Beyond the multiprocessing and batched processing strategies examined earlier, multithreading further reduces processing time. This OP benefits from multithreading due to its intensive I/O procedures (reading images), which balance I/O and CPU utilization. Most other multimodal OPs also require heavy data read/write operations, thus benefiting from the multithreading strategy.

## I  High-Resolution Figures in the Paper

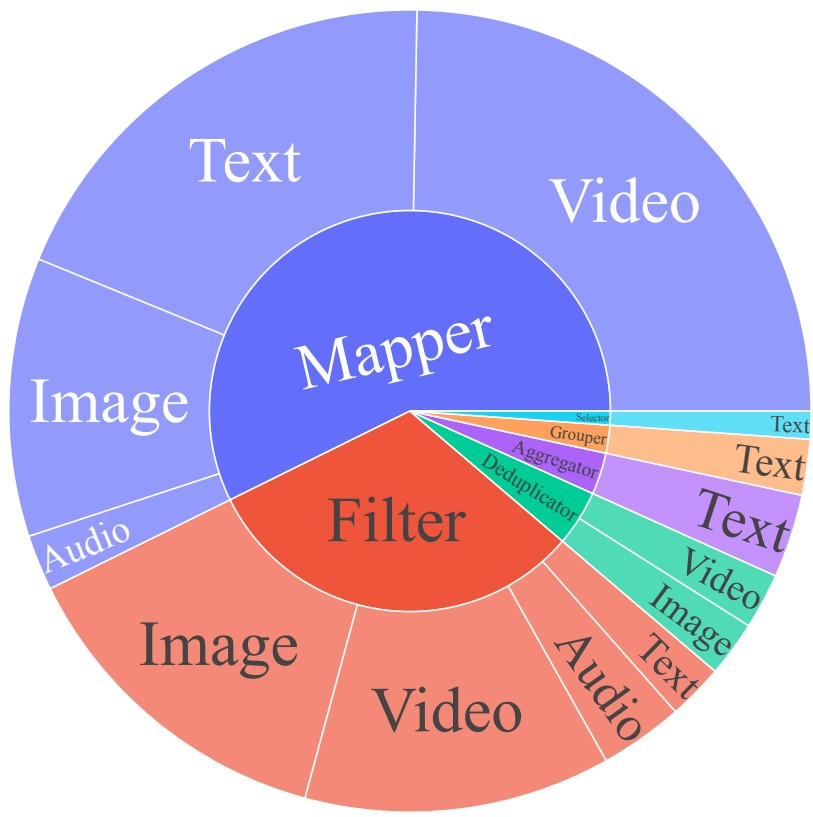

Figure 11: High-resolution version of Figure 2a. The smallest sector in the inner circle is "Selector".

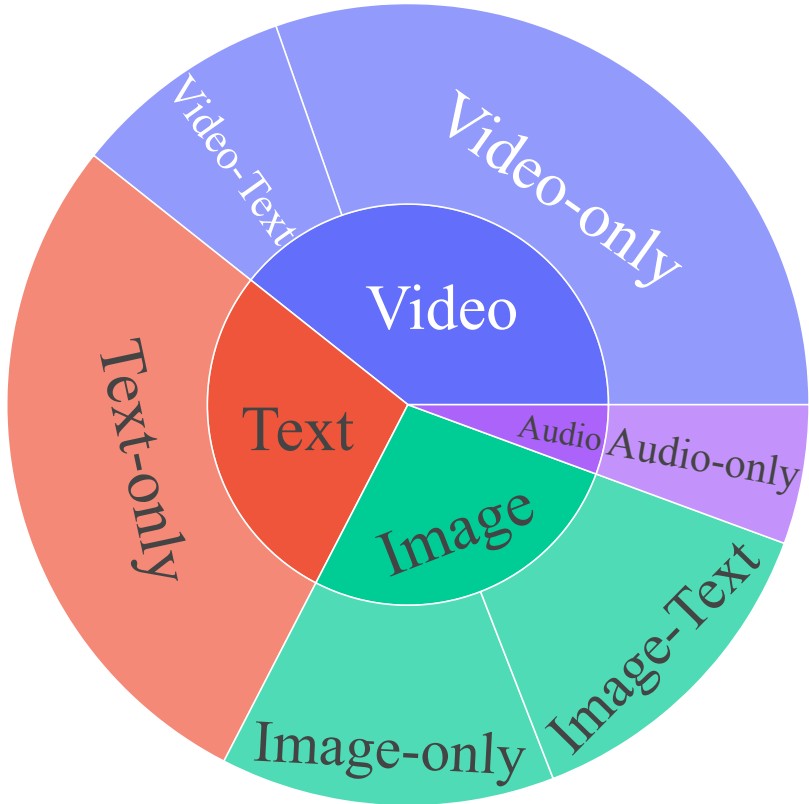

Figure 12: High-resolution version of Figure 2b.

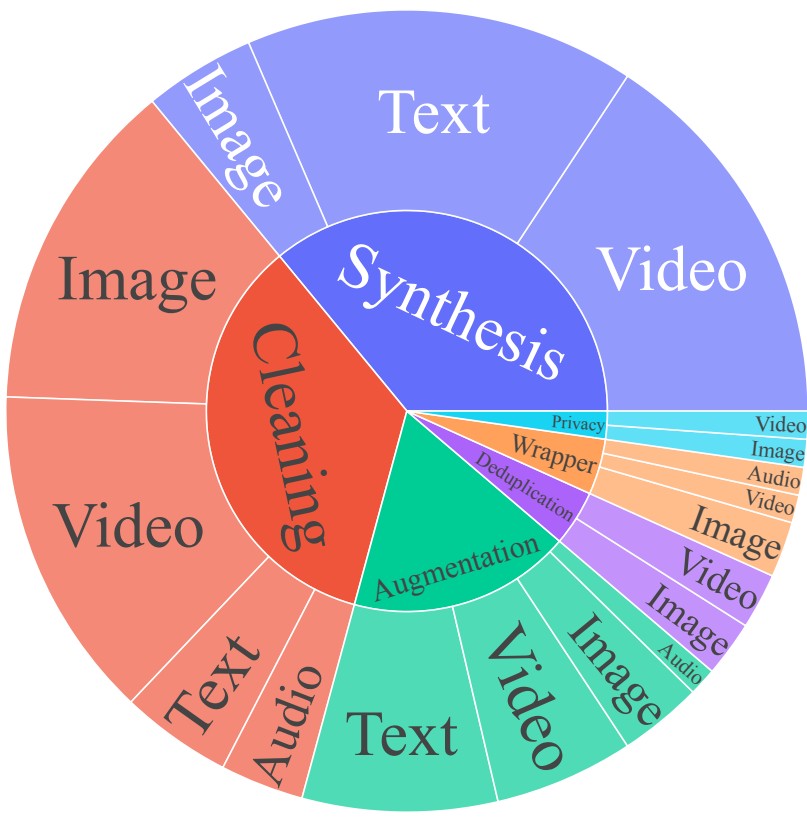

Figure 13: High-resolution version of Figure 2c.

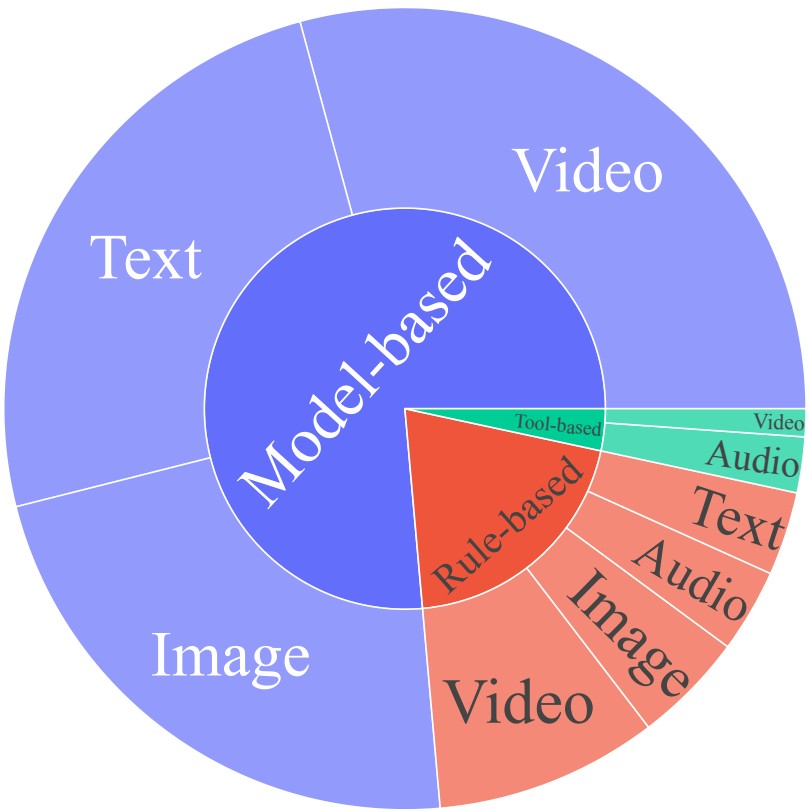

Figure 14: High-resolution version of Figure 2d.

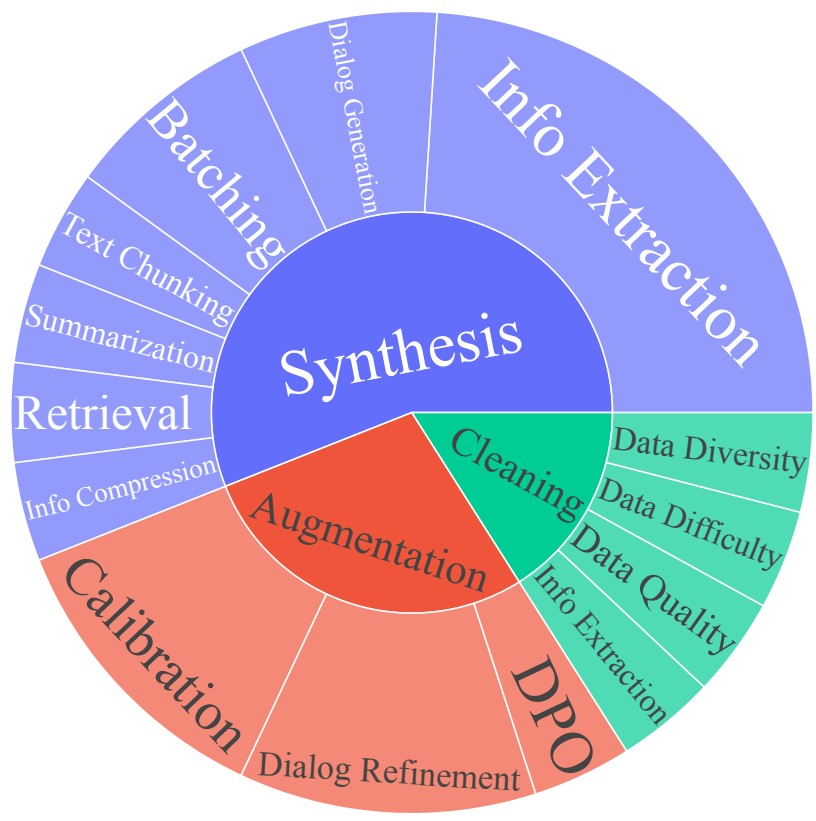

Figure 15: High-resolution version of Figure 3.

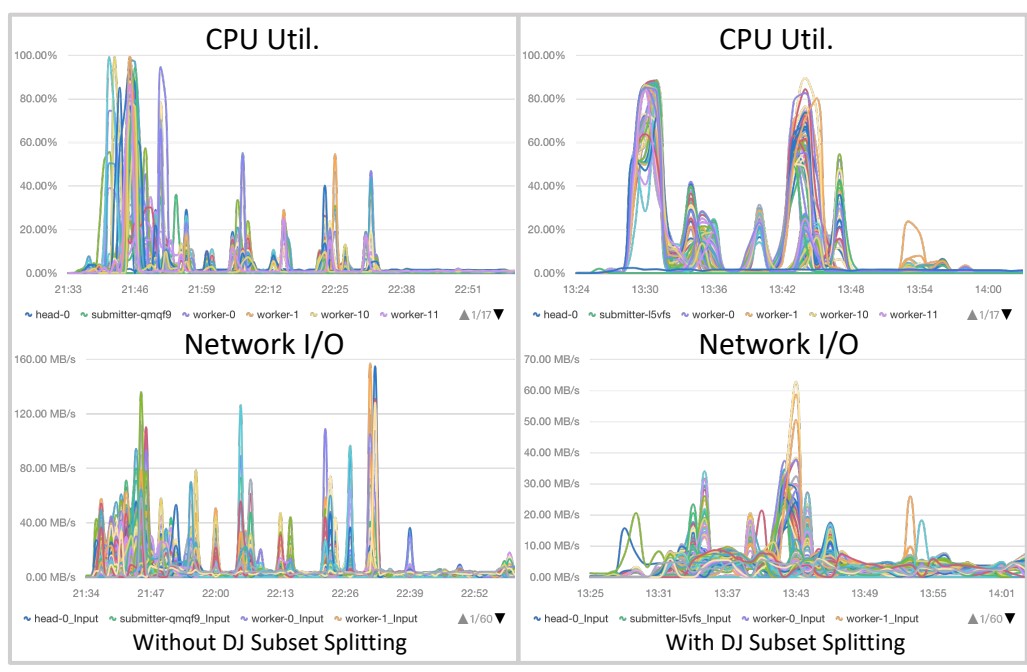

Figure 16: High-resolution version of Figure 4f.

