# OpenReview forum: "Data-Juicer 2.0: Cloud-Scale Adaptive Data Processing for and with Foundation Models"
_NeurIPS.cc/2025/Datasets_and_Benchmarks_Track — NeurIPS 2025 Datasets and Benchmarks Track spotlight_

### Official Review · Reviewer_Q2iW · 2025-06-12

**Rating:** 6
**Confidence:** 4

**Summary:**

This paper presents DataJuicer-2.0 (in the following DJ2), a comprehensive and highly versatile framework for cloud-scale data processing, particularly in the context of machine-learning. Building upon a previous, primarily text-oriented version (DJ1), DJ2 adds functionality for various further data modalities (video, audio, ...) and their multifarious processing, for integrating manifold foundational models, for automated optimization, etc. These functionalities are mostly implemented through flexibly combinable operators (OPs). Starting with a high-level summary of DJ2 and its advancements over DJ1 and a high-level architecture overview, the paper then elaborates on the different available operator types (summarizing them along different dimensions), presents DJ2's access / (system) integration mechanisms (from REST APIs to natural language agents), and outlines its capabilities for flexible and efficient operation / execution, comprising aspects such as OP composition and the orchestration of workflow executions in cloud-scale environments. After this (still summarized) overview, experiments are conducted and reported on in three different scales

**Dataset Code Accessibility:**

Yes

**Ethical Considerations:**

No, there are no or only very minor ethics concerns

**Final Justification:**

Thanks a lot for taking the remarks into account so seriously! Yes, the proposed changes -- if actually implemented -- will properly address my concerns and significantly increase the paper's quality. Especially the re-structuring regarding background & related as well as elevating the "why" work is much, much appreciated.

Based on this and expecting the actual implementation of these changes, I raised my assessment.

**Limitations Weaknesses:**

- Basically all figures are too small to be read without unreasonable zoom - this should definitely be fixed
- The papers writing style is quite uncommon, basically leaving out any introduction, motivation, ... Instead, it immediately jumps into "hey, we have a system and it has the following properties/characteristics". Related work is also provided / discussed in a rather uncommon fashion.
- In a somewhat similar vein, the functionalities presented on are impressive (especially in their count and sophisticatedness), but the whole paper - at least before the experiments – basically lack any scientific reasoning, concepts, considerations, ... In the end, the paper mainly lists the manifold functionalities of DJ2, but mostly without providing any details on the "how" and "why" or on challenges that had to be solved etc. (In fact, challenges, goals, implementation details, etc. are hidden in the appendix, but this is not where a reader would expect it. In the current form, the "actual" paper itself would just be a teaser for the extensive appendix - rather unlucky, in my view.)
- The linkage to "societal impacts" because of OPs for removing NSFW-content does not stick. Don't force it - just having a technical contribution is fine.

These weaknesses are, however, not heavy enough to outweigh the strengths and the benefits from having DJ2 being presented and discussed.

**Strengths Contributions:**

- the paper presents a comprehensive, completely implemented, and impressively mature system for flexible, and extendable large-scale data processing.
- The presented system is solidly evaluated through carefully designed and executed technical experiments (including ablation studies), providing well-founded insights and implications
- The paper is well-written and the presentation ensures it can be followed easily.
- Code is completely open source and extensive supplemental material is included, making everything transparent and reproducible.

---

> ### Author Rebuttal · Authors · 2025-07-30
>
> We sincerely thank the reviewer for their positive and detailed assessment! We are very encouraged by the recognition of our work as a "comprehensive, completely implemented, and impressively mature system" with "solidly evaluated" experiments. The constructive feedback is highly valuable, and we have a clear plan to incorporate it into the final version.
>
>
> ---
>
> **(W1) Improving Figure Readability**
> > **(Fig resolutions)** "Basically all figures are too small to be read without unreasonable zoom - this should definitely be fixed"
>
> Thank you. We agree and will improve the readability of all figures.
>
> *   We will **increase font sizes and redesign layouts** for figures in the main text (e.g., Fig 1, 2, 3) to ensure clarity.
> *   For dense figures like `Figure 4.f`, we will add **high-resolution, vectorized versions to the appendix** and include a direct reference in the main-text captions. This ensures all details are accessible without compromising the main paper's flow.
>
>
> ---
>
> **(W2 & W3) Strengthening the Scientific Narrative & Motivation**
> > **(The writing style)** "The papers writing style is quite uncommon, basically leaving out any introduction, motivation... Instead, it immediately jumps into 'hey, we have a system...'. Related work is also provided / discussed in a rather uncommon fashion... the paper mainly lists the manifold functionalities... without providing any details on the 'how' and 'why' or on challenges..."
>
>
> We are grateful for this constructive feedback on the paper's narrative structure! Our decision to move motivation and design rationale to the appendix was a difficult compromise due to the strict page limit.
> **Thanks to the additional page granted to accepted papers**, we will perform a focused reorganization to elevate the 'why' and 'how' of our work. This is not about adding new experimental results, but about **restructuring existing content** from the appendix and experiments section into a more compelling and logical narrative flow at the beginning of the paper.
>
> 1.  **Elevating the "Why"**: We will restructure the paper to better integrate our analysis of system performance and design choices. Instead of being confined to the experiments section, key insights will be brought forward. These are rooted in:
>     *   **Macro-level analysis**: We will better highlight our discussion on end-to-end performance when scaling from 1x to 125,000x, linking the observed patterns to system design challenges as detailed in *lines 254-262, 271-278, 291-308, and 324-332*.
>     *   **Micro-level analysis**: Key findings from our ablation studies will be more clearly linked to the design challenges they solve in the respective technical sections, including:
>     *   CPU and network resource utilization challenge and analysis (**Figure 4.f**).
>     *   The benefits of automatic workload-aware OP reordering (**Figure 9**).
>     *   The efficiency of automatic GPU resource allocation across different VRAMs and parallelism settings (**Table 3**).
>     *   The impact of batch size and concurrency on batched data processing (**Figure 10.a & 10.b**).
>
> 2.  **Revising the Background & Related Work**: To provide better context upfront, we will restructure the beginning of the paper:
> *   The current `Section 6. Related Work` section will be removed (-0.5 pages).
> *   Instead, we will introduce a new **`Section 2: Preliminaries and Design Rationale`**. This section will be constructed by reorganizing existing content from Appendix A and Section 6 into a more logical flow:
>     *   `2.1 Related Work & Core Challenges` (from Appendix A.1)
>     *   `2.2 System Design Goals` (from Appendix A.2)
>     *   `2.3 Key Differences from Prior Systems` (synthesized from Section 6, Appendix A.) This new section will also feature the below summary table comparing Data-Juicer 1.0 and 2.0, providing readers with a clear, upfront understanding of our core contributions.
>
> | Feature                 | Data-Juicer 1.0                | Data-Juicer 2.0 (New Contributions)                                                                                                                                  |
> | ----------------------- | ------------------------------ | ------------------------------------------------------------------------------------------------------------------------------------------------------------------- |
> | **Data Modality**       | Text-only (~50 OPs)            | + Image, video, audio (~100 new cross-modal OPs)                                                                                                                    |
> | **Operator Types**      | Formatter, Filter, Mapper, etc.  | + Grouper, Aggregator, FusedOP, HumanOP                                                                                                                |
> | **Deduplication**       | Standalone-only                | + Ray-based distributed deduplication                                                                                                                               |
> | **Interaction Methods** | CLI, Low-level APIs            | + RESTful APIs, Web UI, Natural Language Interface                                                                                                                  |
> | **Execution Engines**   | HF Datasets, Ray, Beam         | + Tighter integration with Alibaba PAI-DLC & MaxCompute                                                                                                             |
> | **System Optimization** | OP fusion, Greedy OP reordering  | + Probe-based OP reordering, Auto GPU allocation, Batched processing, OP-wise parallelism, OP insight mining, Streaming I/O, Advanced Fault Tolerance                 |
> | **Compute Scale**       | 1,000+ cores                   | 10,000+ cores                                                                                                                                                       |
> | **Data Scale**          | ~70M samples (TB-level)        | ~70B samples (PB-level)                                                                                                                                             |
> | **Code Contribution**   | 26 PRs, ~34k lines             | + Over 100 new PRs, `~40k` lines added, `~8k` lines deleted                                                                                                         |
>
>
> We hope these structural adjustments can address the reviewer's concern by making the paper's motivations and scientific reasoning prominent from the outset.
>
> ---
>
> **(W4 ) Refining the Discussion on Societal Impact**
> > **(Unsutableness of the linkage to "societal impacts")** "The linkage to 'societal impacts' because of OPs for removing NSFW-content does not stick. Don't force it - just having a technical contribution is fine."
>
> Thank you for the guidance on precision. We agree the original example was not ideal. Our intent was to highlight that our framework includes operators pertinent to data governance, such as privacy and security, which have direct societal relevance.
>
> To address this, we will revise the sentence to be more objective and grounded in the system's capabilities:
>
> > **Revised sentence**: *"Looking ahead, enhancing the framework to cover a broader range of data governance and safety considerations, such as ensuring data privacy across multiple security protocols, remains a key direction for future development."*
>
> We hope this revision can remove the less convincing example and more accurately reflects the system's potential for positive societal impact.
>
> ---
>
> **Closing Remark**
>
> Once again, we thank the reviewer for their time and thoughtful comments! We believe that these these targeted, evidence-based revisions will further enhance the paper's quality and impact. We hope the reviewer finds our response useful and will continue to support our work.

---

### Official Review · Reviewer_cnBx · 2025-06-29

**Rating:** 6
**Confidence:** 5

**Summary:**

The paper present a system called Data-Juicer 2.0, an opensource tool to prepare data for machine learning usage. It is architectured in operators (mappers, filters, dedup) that process pieces of data. It is meant to be an improved of the previous version of the system by providing multimodal operators. The authors do evaluations by varying the number of samples (up to 70B text samples).

**Additional Feedback:**

Thank you for building this tool, and providing a detailed technical report on how it was built and why!
I have some follow up questions:
1. Was this tool used to process large scale multimodal datasets? (multi billions image datasets; hundred of thousands of hours of audio or video) If not, do you have plans to do it, and what adaption to the tools would make it possible?
2. Was this system used to produce datasets that were used successfully to produce high quality models?
3. The presented system has a lot of functionalities which means a high maintenance cost. If you were to produce a minimal version of this system; what would it contain?

**Dataset Code Accessibility:**

Yes

**Dataset Code Comments:**

The code is available on github and already used by many users. It is well documented.

**Ethical Comments:**

No ethical concerns

**Ethical Considerations:**

No, there are no or only very minor ethics concerns

**Final Justification:**

After reading the rebuttal, I am upgrading my recommendation to Strong Accept (6). The authors have convincingly addressed all key concerns, and the system's maturity, impact, and extensibility are now even clearer. My updated assessment is based on the following:

Resolved concern: Large-scale multimodal evaluation
The authors clarified that they processed 1.4B image-text pairs in under 2 hours using 3200 cores. This confirms that the system already operates at billion-scale multimodal levels, as previously unclear.

Resolved concern: Real-world usage and model training impact
Examples from Alibaba’s foundation models, ICML/VBench, and open-source datasets demonstrate the tool’s role in real-world high-performance model development. This shows both academic and industrial relevance.

Resolved concern: Maintainability
The planned modular split into separate packages (e.g., datajuicer-core, datajuicer-text) directly addresses concerns about usability and long-term maintainability.

Given the system’s scalability, real-world adoption, extensibility, and clear roadmap, I now believe it will have high impact across both academic and applied ML communities.

**Limitations Weaknesses:**

The main weakness of this paper is the scaling evaluation. Although the paper claims to evaluate up to 70B samples, it is clear the author mean 70B text samples.
Figure 4 confirms this by mentioning only medium-scale multimodal data.
Handling large scale (many billions) of image or video data points introduces significant constraints requiring to re-design the systems to avoid copies and select the appropriate resources during processing (network vs cpu)

The paper would benefit a lot from evaluating on large multimodal datasets (image and video) which may show that the system can indeed be used for multimodal processing at scale.

As a separate point, the paper does not mention how this tool was used to train large models and how functionalities of the tools specifically help reach better results either in iteration speed or quality results.

**Strengths Contributions:**

The architecture of the proposed system has a lot of benefits compared to generic data processing tools: it provides operators that work on the most common modalities of datasets used for machine learning. It has support for several executor backends (ray data; hugging face dataset ; local). It also provides multiple interfaces for users to interact with the system (rest; web app ; llm ; python)

The paper clearly explains the architecture of this system with figures that make the architecture and modality support clear.

The evaluation section helps understand the performance profiles (network; cpu) as the system scales.

Having an HumanOP is a novel feature that may help build dataset at small scale and integrate humans for high complexity datasets.

---

> ### Author Rebuttal · Authors · 2025-07-30
>
> We are sincerely grateful to the reviewer for their strong support and their insightful, detailed feedback! Their expertise is evident, and their questions help us clarify the most impactful aspects of our work. We are pleased to address each point below. For brevity, we denote paragraphs in weaknesses as *W* and Additional Feedback as *F*.
>
> ---
>
> ### **(W1, W2 & F1) On Large-Scale Multimodal Evaluation and Future Scaling**
>
> >  *"The main weakness... is the scaling evaluation... Was this tool used to process large scale multimodal datasets? (multi billions image datasets...)... what adaption to the tools would make it possible?”*
>
> Thank you for this crucial question. We want to confirm that **we have indeed evaluated Data-Juicer on billion-scale multimodal data**, and we apologize that the magnitude of these experiments was not sufficiently highlighted in the submitted version.
>
> **1. Clarification of Existing Billion-Scale Experiments:**
>
> The scale of our multimodal experiments was perhaps obscured by our use of relative multipliers. We will make this explicit in the final version.
> *   In **lines 283-287**, the "2500x" experiment corresponds to processing **1.4 billion text-image pairs**.
> *   Our system processed this dataset in **under 2 hours** using 3200 Ray-DLC cores, a result that demonstrates strong real-world scalability.
> *   We will revise the main text and captions to feature these absolute sample counts prominently, ensuring the scale of our evaluation is immediately clear.
>
> **2. A Roadmap for Petabyte-Scale Adaptation:**
>
> You are absolutely right that scaling to the next order of magnitude requires forward-thinking architectural design. We are excited to share our roadmap and actively designing a next-generation **pipeline optimization framework** to meet this challenge. This goes beyond simply adding more cores and addresses fundamental bottlenecks:
>
> *   **Advanced Fusion Strategies:** Building on our existing filter fusion, we are adding `MapperFusion`. This will analyze the pipeline's Abstract Syntax Tree (PipeAST) and merge consecutive, computationally compatible OP operations (e.g., image resizing followed by cropping) into a single, fused kernel, aiming to further reduce intermediate data materialization and I/O.
>
> *   **Intelligent Column Pruning:** For wide-schema datasets (e.g., Parquet files with hundreds of columns, especially in scenarios involving a large number of raw signals like self-driving), loading all data is wasteful if an operator only needs a few columns. Our planned `ColumnPruner` will analyze the entire pipeline's data dependencies and selectively load only the required columns from storage from the very beginning.
>
> *   **Operator Pushdown:** We are designing a mechanism to "push down" inexpensive filtering operations to the earliest possible stage. For example, a `Filter` that checks image metadata (like resolution or file size) can be applied *during* the initial data listing phase, *before* the full, potentially large, image files are even downloaded or deserialized. This avoids immense I/O and compute for samples that would be discarded anyway.
>
> *   **Fine-grained Resource Tuning:** We plan to introduce a `ResourceTuner` that moves beyond static resource allocation. It will monitor real-time resource usage (CPU, GPU VRAM, memory bandwidth) and leverage new features in execution engines like Ray (e.g., compiled graphs) to dynamically adjust parallelism and resource assignment on a per-operator basis, ensuring optimal system utilization.
>
> We will add a dedicated discussion of these billion-scale results and our roadmap for future scaling to the final version.
>
> ---
>
> ### **(W3 & F2) On Real-World Impact and Use in Model Training**
>
> > *"the paper does not mention how this tool was used to train large models and how functionalities... help reach better results either in iteration speed or quality results."*
>
> This is an excellent question about the system's tangible impact. Data-Juicer is not just an open-source prototype; it has been instrumental in producing several high-quality models and datasets across industry and academia.
>
> *   **State-of-the-Art Model Development:** Data-Juicer is the direct enabler of these successes. For example, in our ICML'24 Spotlight work, the ability to rapidly iterate on complex data processing pipelines using Data-Juicer's Sandbox was the **key factor** that led to a model achieving a **top rank on VBench**. This directly answers the question of how our system's functionalities lead to 'better quality results' and dramatically improve 'iteration speed'.
>
> *   **Powering Enterprise Foundation Models:** Data-Juicer is the industrial-strength backbone for data processing for numerous **enterprise-level foundation models** on Alibaba Tongyi and Alibaba Cloud PAI. It has been battle-tested on petabyte-scale proprietary workloads for over a year, proving its robustness, scalability, and value in production environments where data quality and iteration speed are paramount.
>
> *   **Enabling Open-Source Research:** As a key enabler for the community, the system has powered data pipelines to improve pre-training datasets (e.g., as shown in our prior v1 work, SIGMOD'24) and generate novel open-source datasets for post-training. For example, those papers mentioned in the github readme page demonstrated diverse use-cases, for contrastive synthesis (ImgDiff, CVPR'25), reasoning enhancement (MindGYM), diversity enhancement (DaaR), and evaluation (HumanVBench, DetailMaster).
>
>
> While this paper's primary focus is the system's architecture and performance, its ultimate value lies in **accelerating the data-model co-development cycle**. By making large-scale, complex data processing faster, cheaper, and more reproducible, Data-Juicer empowers researchers to conduct more experiments and iterate on data strategies, which is often the key to unlocking better model performance.
>
> ---
>
> ### **(F3) On System Maintenance and a "Minimal Version"**
>
> > *"The presented system has a lot of functionalities which means a high maintenance cost. If you were to produce a minimal version... what would it contain?"*
>
> We wholeheartedly agree that for a feature-rich system, maintainability and a low barrier to entry are critical for broad adoption. To address this and provide users with a "minimal" experience tailored to their needs, we are undertaking a strategic, forward-compatible refactoring.
>
> Our plan is to restructure Data-Juicer from a monolithic repository into a *GitHub organization* composed of modular, interconnected packages. This allows users to install and understand only the components they need, drastically reducing dependency overhead and cognitive load. A user could, for example, achieve a minimal text-processing setup by simply installing `datajuicer-text` (which implicitly requires `datajuicer-core`).
>
> | New Repository      | Description                                          | Corresponding Modules in Current Library                          |
> | ------------------- | ---------------------------------------------------- | ----------------------------------------------------------------- |
> | `datajuicer-core`   | The core abstractions, execution engine, and I/O.    | `core/`, `executor/`, `format/`, `config/`, `download/`           |
> | `datajuicer-text`   | Operators and recipes for large language models.     | `ops/`, `tools/`, `configs/` tailored to text scenarios           |
> | `datajuicer-mm`     | Operators and recipes for vision/multimodal models. | `ops/`, `tools/`, `configs/` tailored to multimodal scenarios    |
> | `datajuicer-sandbox`| A middleware for data-model co-development.          | `core/sandbox/`                                                   |
> | `datajuicer-agent`  | The natural-language and visual interface.           | `analyzer/`, `monitor/`, `service/`, `agent` branch                 |
> | more potential repo | ... | |
>
> This modularization directly serves the goal of a "minimal version" by allowing users to self-select their desired feature set. It also complements our existing "DJ-lite" optimizations, such as **lazy-loading of heavy operator dependencies** and the use of the efficient **`uv` package manager** (lines 147-152), which will be preserved and enhanced in the new structure.
>
> Moreover, we hope it can also **lower the barrier for community contributions**, allowing experts to develop specialized operators (e.g., datajuicer-bio) without needing to master the entire codebase. This fosters a sustainable and thriving open-source ecosystem, a core goal of our project.
>
> ---
>
> We are deeply grateful for your strong support and expert engagement with our work. We believe these detailed clarifications and our planned improvements will make the paper an even stronger contribution. Thank you again for your positive and encouraging assessment!

---

> > ### Comment · Reviewer_cnBx · 2025-08-08
> > **Answer to rebuttal**
> >
> > Thank you for the detailed rebuttal. Your clarifications on the 1.4B image-text pair experiment and upcoming scaling features (MapperFusion, ColumnPruning, etc.) address my concerns on large-scale multimodal processing.
> >
> > The examples showing Data-Juicer’s use in production (e.g., Tongyi, VBench) clearly demonstrate real-world impact.
> >
> > The planned modular split into core packages is a great step toward reducing maintenance and improving usability.
> >
> > Based on this I will update the score for the review.

---

> > > ### Author Response · Authors · 2025-08-09
> > > **Thanks for your response and support**
> > >
> > > Dear Reviewer cnBx,
> > >
> > > Thank you so much for your thoughtful and positive reply!
> > >
> > > We were already very grateful for your *strong initial support* for our paper. To hear that our detailed rebuttal has further addressed your concerns and earned your *even stronger endorsement* is largely rewarding for our team.
> > >
> > > Your recognition of our work on large-scale multimodal processing, its real-world impact, and our future-oriented modular architecture means a great deal to us.
> > >
> > > We sincerely appreciate your constructive engagement throughout this process and your willingness to update your assessment. We are more motivated than ever to continue advancing Data-Juicer as a more valuable resource for the community. Thank you once again.

---

### Official Review · Reviewer_5StA · 2025-07-02

**Rating:** 5
**Confidence:** 2

**Summary:**

This paper introduces Data-Juicer 2.0, a comprehensive system featuring 100+ operators for processing text, image, video, and audio data, supporting tasks like analysis, synthesis, annotation, and model post-training. With seamless integration for platforms like Hugging Face and Ray, it enhances usability, efficiency, and programmability. Evaluations show its scalability, efficiently processing TB-level data with 10k+ CPU cores. Open-sourced and adopted in research and industry (Alibaba Cloud).

**Dataset Code Accessibility:**

Yes

**Dataset Code Comments:**

The submitted datasets and codes are easily accessible and well-documented. The author also provides comprehensive technical details for reproduction.

**Ethical Comments:**

No significant ethical concerns remain in this paper.

**Ethical Considerations:**

No, there are no or only very minor ethics concerns

**Final Justification:**

After the rebuttal, most of my concerns are addressed, so I raise the score to accept.

**Limitations Weaknesses:**

1. The proposed system provides a toolkit and process for data processing, but as a submission to the "dataset & benchmark track", it lacks the contribution of new standard datasets/evaluation benchmarks. I'm not very familiar with the DB track, I'm not sure whether this will be a problem.
2. The novelty of this paper is not clear.

**Strengths Contributions:**

1. The proposed Data-Juicer 2.0 has been widely implemented and maintained in real industries and academic applications (Alibaba Cloud PAI products).
2. It supports mainstream engines such as HuggingFace, Ray, and MaxCompute
3. The paper reports detailed experiments on different scales and different engines, demonstrating good scalability and efficiency.
4. Data-Juicer 2.0 launches many multimodal operators to support complex foundation model data pipelines.

---

> ### Author Rebuttal · Authors · 2025-07-30
>
> We sincerely thank the reviewer for their thoughtful feedback and supportive rating! We particularly appreciate their candor regarding the scope of the Datasets & Benchmarks track, and we are happy to provide clarifications that we believe will address their primary concerns.
>
> ---
>
> ### **W1: On Track Alignment and Contribution Type**
>
> > *"The proposed system provides a toolkit..., but as a submission to the "dataset & benchmark track", it lacks the contribution of new standard datasets/evaluation benchmarks. I'm not very familiar with the DB track, I'm not sure whether this will be a problem."*
>
> We thank the reviewer for raising this important point. You are correct that our primary contribution is a **system/tool**, not a new dataset. We chose this track specifically because its Call for Papers (CFP) explicitly encourages and values such contributions.
>
> Our work directly aligns with the track's stated scope in several ways:
>
> *   **Our chosen "Primary Area"** in the submission system was: `"AL/ML data processing and benchmarking Infrastructure (e.g., ... distributed data processing solutions, scalable data analysis)"`.
>
> *   The NeurIPS 2025 D&B CFP states it aims to support the open-source movement by encouraging:
>     > *"submissions of **open-source libraries and tools** that enable or accelerate ML research."*
>
> *   The CFP further defines the track's scope to include:
>     > *"**Data-centric AI methods and tools**, e.g. to measure and improve data quality or utility, or studies in data-centric AI that bring important new insight."*
>
> *   And it also includes:
>     > *"**Advanced practices in data collection and curation** that are of general interest even if the data itself cannot be shared."*
>
> Given these explicit guidelines, we believe our work is not only a strong fit but also exemplifies the kind of open-source infrastructure contribution the track aims to champion. We hope this clarification helps the reviewer assess our work with greater confidence in its alignment and contribution.
>
> ---
>
> ### **W2: On Novelty and Technical Contributions**
>
> > *"The novelty of this paper is not clear."*
>
> Thank you for this feedback. We acknowledge that our novel contributions, while discussed throughout the paper (e.g., lines 54-81, 110-113, 364-372), could be presented more clearly.
>
> #### 1. From the technical novelty perspective
>
> To provide a consolidated and structured overview, we have created a table below that explicitly highlights the significant advancements of Data-Juicer 2.0 over its predecessor. We will add this table to the final version to make our novel contributions immediately apparent to all readers.
>
> | Feature                 | Data-Juicer 1.0                | Data-Juicer 2.0 (**New Contributions**)                                                                                                                             |
> | ----------------------- | ------------------------------ | ------------------------------------------------------------------------------------------------------------------------------------------------------------------- |
> | **Data Modality**       | Text-only (~50 OPs)            | + Image, video, audio (~100 new cross-modal OPs)                                                                                                                    |
> | **Operator Types**      | Formatter, Filter, Mapper, etc.  | + Grouper, Aggregator, FusedOP, Human-in-the-loop OP                                                                                                                |
> | **Deduplication**       | Standalone-only                | + Ray-based distributed deduplication                                                                                                                               |
> | **Interaction Methods** | CLI, Low-level APIs            | + RESTful APIs, Web UI, Natural Language Interface                                                                                                                  |
> | **Execution Engines**   | HF Datasets, Ray, Beam         | + Tighter integration with Alibaba PAI-DLC & MaxCompute                                                                                                             |
> | **System Optimization** | OP fusion, Greedy OP reordering  | + Probe-based OP reordering, Auto GPU allocation, Batched processing, OP-wise parallelism, OP insight mining, Streaming I/O, Advanced Fault Tolerance                 |
> | **Compute Scale**       | 1,000+ cores                   | 10,000+ cores                                                                                                                                                       |
> | **Data Scale**          | ~70M samples (TB-level)        | ~70B samples (PB-level)                                                                                                                                             |
> | **Code Contribution**   | 26 PRs, ~34k lines             | + Over 100 new PRs, `~40k` lines added, `~8k` lines deleted
>
> As this table demonstrates, Data-Juicer 2.0 represents a notable leap in modality support, architectural sophistication, optimization techniques, and large scales, establishing its significant novelty.
>
> #### 2. From the innovation enabler perspective
>
> Moreover, as an open-sourced **infra**, Data-Juicer has been instrumental in producing novel high-quality models and datasets across industry and academia.
>
> *   **State-of-the-Art Model Development:** Data-Juicer is the direct enabler of many innovations. For example, in our ICML'24 Spotlight work, the ability to rapidly iterate on complex data processing pipelines using Data-Juicer's Sandbox was the **key factor** that led to a model achieving a **top rank on VBench**. This directly answers the question of how our system's functionalities lead to 'better quality results' and dramatically improve 'iteration speed'.
>
> *   **Powering Enterprise Foundation Models:** Data-Juicer is the industrial-strength backbone for data processing for numerous **enterprise-level foundation models** on Alibaba Tongyi and Alibaba Cloud PAI. It has been battle-tested on petabyte-scale proprietary workloads for over a year, proving its robustness, scalability, and value in production environments where data quality and iteration speed are paramount.
>
> *   **Enabling New Open-Source Research:** As a key enabler for the community, the system has powered data pipelines to improve pre-training datasets (e.g., as shown in our prior v1 work, SIGMOD'24) and generate novel open-source datasets for post-training. For example, those papers mentioned in the github readme page demonstrated diverse use-cases, for contrastive synthesis (ImgDiff, CVPR'25), reasoning enhancement (MindGYM), diversity enhancement (DaaR), and evaluation (HumanVBench, DetailMaster).
>
> While this paper's primary focus is the system's architecture and performance, its ultimate value lies in **accelerating the data-model co-development cycle**. By making large-scale, complex data processing faster, cheaper, and more reproducible, Data-Juicer empowers researchers to conduct more experiments and iterate on data strategies, which is often the key to unlocking better model/data innovations.
>
> ---
>
> ### Closing Remark
>
> We are grateful for the opportunity to improve our work based on this feedback. Given their low confidence, we hope that these clarifications can directly address the reviewer's concerns regarding track fit and novelty, and provides the necessary context to better assess our paper. Feel free to contact us if you have any further inquiries or recommendations. We appreciate your time and feedback immensely.

---

> > ### Comment · Reviewer_5StA · 2025-08-05
> >
> > Thanks for the detailed rebuttal, most of my concerns are addressed. I will raise my score to accept.

---

> > > ### Author Response · Authors · 2025-08-06
> > > **Thank you for your constructive engagement!**
> > >
> > > Dear Reviewer 5StA,
> > >
> > > Thank you so much for your response and positive feedback! We are very encouraged that our rebuttal was helpful, and we truly appreciate your support for our work and your decision to raise the score.
> > >
> > > In light of the chairs' recent reminder about the author-reviewer discussion period and their encouragement for active engagement, we just wanted to add that *we are standing by*. We hope our rebuttal is helpful to the entire review team and are ready to clarify further points.
> > >
> > > Thank you again for your valuable time and re-evaluation!

---

### Official Review · Reviewer_m1C3 · 2025-07-04

**Rating:** 4
**Confidence:** 2

**Summary:**

Data-Juicer 2.0 addresses the inefficiency of traditional frameworks in processing multimodal data for foundation models by introducing a unified system with 100+ cross-modal operators, optimized compatibility with popular platforms, and adaptive runtime management. It achieves scalable TB-level data processing with 10k+ CPU cores, demonstrating practical success in applications like Alibaba Cloud PAI while offering user-friendly interfaces for broader adoption.

**Dataset Code Accessibility:**

Yes

**Ethical Considerations:**

No, there are no or only very minor ethics concerns

**Limitations Weaknesses:**

1. Provide a more thorough discussion of the robustness, and potential applications beyond Alibaba Cloud PAI.
2. I recommend thoroughly revising this paper, improving the clarity and the reader’s understanding.
3. Exploring the reasons behind the success of these techniques and providing intuitive explanations would contribute to the overall scientific contribution of the work. For example, the main experiments are conducted on Alibaba Cloud resources, the author should discuss the limitations of Alibaba Cloud resources, and conduct the validation on more Foundation Models.
4. In the main text, the author should provide more attention to the experimental validation and analysis of Data Jiucer 2.0.
5. Figure 1 is not clear. The same issue can be found in Figure 4.
6. What are the main contributions of Data Jiucer 2.0 compared to Data Jiucer?
7. In the provided dataset link, https://github.com/modelscope/data-juicer, the author has provided many published papers. What is the relationship between these papers and the dataset used in this paper?

**Strengths Contributions:**

1. The presentation of the paper is good
2. The issue of traditional data processing frameworks is important.
3. This paper is well-written.
4. The proposed Data-Juicer 2.0, is superior to the previous method.
5. This paper provides some experiments, validating the statements proposed.

---

> ### Author Rebuttal · Authors · 2025-07-30
>
> We thank the reviewer for the valuable comments, which helped us improve the paper! We address each point below and respectfully hope these responses warrant a more positive evaluation.
>
> ---
>
> ### **W1: On Robustness and Broader Applications**
>
> > *"Provide a more thorough discussion of the robustness, and potential applications beyond Alibaba Cloud PAI.”*
>
> Thank you! We agree these are crucial.
>
> **1. On Robustness:**
>
> Beyond the sample-level tolerance discussed in Section F.2, we have recently architected system-level robustness to complement Ray's native capabilities:
>
> *   **LLM OP Resilience:** LLM operators can return schema-violating outputs (e.g., malformed `dict`), causing `TypeError` during aggregation. Our system adds *pre-flight checks* for LLM responses, validating schema and latency, and *automatic retries* with backoff for failed or non-compliant calls.
>
> *   **Advanced Fault Tolerance & Recovery:** Ray’s pipeline-level fault tolerance can be coarse. A single task failure requires a full restart. To improve this, we are implementing a more granular recovery mechanism (via our public PR #748) which introduces:
>     *   *Partitioning:* Adaptive and size-based data partitioning to manage memory and enable parallel recovery.
>     *   *Checkpointing:* OP-level checkpointing after each stage, with support for multiple formats (Parquet, Arrow, JSONL) and automatic cleanup.
>     *   *Event Logging:* Real-time monitoring of all operations, partitions, and system events for fine-grained observability and debugging.
>     *   *Fine-grained Recovery:* Multiple recovery strategies (e.g., restart from checkpoint, graceful degradation) to ensure progress even if some partitions fail.
>
> **2. On Broader Applications:**
>
> We wish to clarify that *Alibaba Cloud PAI* serves as a gateway to a diverse user base, providing us with invaluable feedback from real-world, large-scale deployments.
> *   **Users:** Our system supports critical workloads for Alibaba's internal teams (e.g., *Tongyi lab*) and numerous external enterprise clients with PB-scale data needs (names withheld for compliance).
> *   **Scenarios:** Data-Juicer is being extended to novel domains such as *spatial-temporal intelligence*, *self-driving*, and complex *multilingual/multi-speaker ASR*.
> *   **Technical Integrations:** We are extending Data-Juicer by integrating with formats like *Lance* for faster queries, enabling *direct fetching from remote storage*, and tightening integration with downstream training frameworks (e.g., by pre-emptively handling tokenization).
>
> We will add these discussions to the final version, to better showcase that Data-Juicer is not a research prototype but a battle-tested, industrial-strength system trusted with mission-critical, petabyte-scale workloads.
>
> ---
>
> ### **W2 & W5: On Paper Clarity and Figure Readability**
>
> > *"I recommend thoroughly revising this paper..."* and *"Figure 1 is not clear... Figure 4."*
>
> Thank you! We will carefully revise it:
>
> -   *Define key terms* on first use.
> -   *Improve figures:* Replace low-res figures (e.g., Fig. 1, Fig. 4.f) with high-res vector graphics, improving layout and font size.
> -   *Add larger versions* of key figures to the appendix.
> -   *Refine language* throughout the paper for precision.
>
> ---
>
> ### **W3 & W4: On Deeper Experimental Analysis**
>
> > *"Exploring the reasons behind the success..."*, *"... more attention to the experimental validation..."*
>
> We agree explaining why our system is efficient is vital. Perhaps it was not highlighted sufficiently and we summarize it below:
>
> **1. Analysis of Performance Gains:**
>
> *   **Macro-level Analysis:** We scaled the dataset from 1x to 12,500x to analyze end-to-end performance, discussing implications at each scale in *lines 254-262, 271-278, 291-308, and 324-332*.
> *   **Micro-level Analysis:** We conducted ablation studies on key optimizations:
>     - *Resource Utilization:* Analyzed in `Fig. 4.f`.
>     - *Workload-aware OP Reordering:* Benefits shown in `Fig. 9`.
>     - *Automatic GPU Resource Allocation:* Performance across different VRAMs and parallel settings detailed in `Table 3`.
>     - *Batched Data Processing:* Impact of batch size and concurrency shown in `Fig. 10`.
>
> The analysis was scattered due to space limits. We are grateful for the positive rating and note that accepted papers are allowed an *additional content page*. We will use this space to consolidate these analyses into more structured and prominent positions, providing a better flow on how our design choices directly translate to performance gains.
>
> **2. On Cloud Resource Limitations:**
>
> Our cloud use was to ensure hardware consistency for reproducibility. We also analyzed the infra's impact. As discussed in *lines 293-308*, we identified the importance of *storage-compute co-design*, showing how high-bandwidth storage and a robust head node are critical—a generalizable insight for large-scale systems.
>
> **3. On Validation with More Foundation Models:**
>
> This submission's primary focus is on the **data processing system's efficiency, scalability, and architecture**. The primary value of Data-Juicer is as an **open-source infra** that accelerates research by making data processing faster and cheaper.
>
> While this paper focuses on the system's architecture, its ultimate validation lies in the **downstream success it enables**. For instance:
> *  Our system was instrumental in Data-Juicer Sandbox (ICML'25 spotlight). This work, which achieved the top rank on the VBench benchmark, is a direct testament to how its efficient and high-quality data processing empowers researchers to build SOTA models.
> *  Our SIGMOD'24 paper details how Data-Juicer was used to build a superior pre-training dataset and model from scratch.
> *   Many other works like those mentioned in code link, have used Data-Juicer for generating high-quality models with better training data (like ImgDiff, CVPR'25), and evaluation data (HumanVBench, DetailMaster).
>
> It empowers researchers to iterate more quickly, thus enabling the creation of better models within limited time and resource budgets.
>
> ---
>
> ### **W6: On Contributions of Data-Juicer 2.0 over 1.0**
>
> > *"What are the main contributions of Data Juicer 2.0 compared to Data Jiucer?"*
>
> We discussed these advancements in several sections (e.g., lines 54-81, 110-113, 364-372). To provide maximum clarity, we summarize the key advancements in the table below, which we will add to the final version.
>
> | Feature                 | Data-Juicer 1.0                | Data-Juicer 2.0 (New Contributions)                                                                                                                                  |
> | ----------------------- | ------------------------------ | ------------------------------------------------------------------------------------------------------------------------------------------------------------------- |
> | **Data Modality**       | Text-only (~50 OPs)            | + Image, video, audio (~100 new cross-modal OPs)                                                                                                                    |
> | **Operator Types**      | Formatter, Filter, Mapper, etc.  | + Grouper, Aggregator, FusedOP, HumanOP                                                                                                                |
> | **Deduplication**       | Standalone-only                | + Ray-based distributed deduplication                                                                                                                               |
> | **Interaction Methods** | CLI, Low-level APIs            | + RESTful APIs, Web UI, Natural Language Interface                                                                                                                  |
> | **Execution Engines**   | HF Datasets, Ray, Beam         | + Tighter integration with Alibaba PAI-DLC & MaxCompute                                                                                                             |
> | **System Optimization** | OP fusion, Greedy OP reordering  | + Probe-based OP reordering, Auto GPU allocation, Batched processing, OP-wise parallelism, OP insight mining, Streaming I/O, Advanced Fault Tolerance                 |
> | **Compute Scale**       | 1,000+ cores                   | 10,000+ cores                                                                                                                                                       |
> | **Data Scale**          | ~70M samples (TB-level)        | ~70B samples (PB-level)                                                                                                                                             |
> | **Code Contribution**   | 26 PRs, ~34k lines             | + Over 100 new PRs, `~40k` lines added, `~8k` lines deleted                                                                                                         |
>
> ---
>
> ### **W7: On the Relation to Other Cited Works**
>
> > *"In the provided dataset link, ... What is the relationship between these papers and the dataset used in this paper?"*
>
> The "Code URL" link in the submission system points to our **open-source system repository**, *not* a new dataset being released with this paper. The relationship is as follows:
>
> - Other papers **use Data-Juicer as a tool** to facilitate their research. For example, they leverage our system's pipelines to generate novel datasets (like ImgDiff) or to train models. Data-Juicer is the *enabling infrastructure* for them.
> - This submission **benchmarks the Data-Juicer system itself**. We use a standard, widely-used dataset (LLaVA's pre-training data, synthetically scaled up) to rigorously measure our system's efficiency and scalability, detailed in *lines 239-241* and *Section H.1*.
>
> ---
>
> We appreciate the opportunity to improve our paper based on this constructive feedback and respectfully hope the reviewer will consider our responses favorably. Thanks again!

---

> > ### Comment · Reviewer_m1C3 · 2025-08-07
> >
> > After reading the author's response, the main concerns are addressed. I still hope the author provides potential applications beyond Alibaba Cloud PAI.

---

> > > ### Author Response · Authors · 2025-08-07
> > > **Thanks for the engagement**
> > >
> > > Dear Reviewer m1C3,
> > >
> > > Thank you very much for your positive feedback and for acknowledging that our previous response addressed your main concerns. We appreciate the opportunity to provide further details on the broad adoption and future potential of Data-Juicer, as you suggested.
> > >
> > > **1. Broad Adoption and Community Integration**
> > >
> > > We wish to clarify that Data-Juicer has already established a diverse user base. The project's value is demonstrated not just by its use in large-scale industrial settings but also by its integration into the wider open-source community.
> > >
> > > *   **Deep Ecosystem Collaboration:** Data-Juicer is actively co-developed with other cutting-edge infrastructure teams. For instance, we maintain an ongoing collaboration with the **NVIDIA** and **Ray team** to ensure mutual optimization. A lead author of Ray recently noted in our official repository (Issue #742): *"This is a really exciting project... We're seeing more users mention Data-Juicer as a key library on top of Ray."* This external validation from other foundational frameworks highlights our growing applications in the AI stack.
> > >
> > > *   **Widespread Industry and Academic Use:** To our knowledge, as an Apache-2.0-license project, Data-Juicer is used by a diverse range and dozens of organizations beyond Alibaba. Our user community, which provides invaluable feedback, includes leading companies in sectors like social media & content platforms (e.g., Xiaohongshu, Ximalaya), e-commerce & finance (e.g., JD.com, Ant Group), consumer electronics (e.g., Xiaomi, OPPO), automotive (e.g., BYD Auto, GAC Group), and numerous AI research labs and universities.
> > >
> > > **2. Expanding on Future and Potential Applications**
> > >
> > > Building on this foundation, Data-Juicer is architected to enable several emerging and future applications, including but not limited to:
> > >
> > > *   **Support of Evolving Agents:** This is a key area of focus. We are enabling the creation of an autonomous *"data flywheel"* for agents: `interaction → generating experience → filtering/synthesizing high-quality data → refining knowledge → guiding better future interactions`. To make this vision concrete, we are actively integrating and co-optimizing with other open-source projects like *AgentScope* and *Trinity-RFT*. This moves beyond simple static data processing to creating a dynamic experience data loop for agent continuous-learning.
> > >
> > > *   **Broader Enterprise and Scientific Use:** The system is well-suited for transforming unstructured enterprise data (e.g., customer reviews, support tickets) into structured insights for *Business Intelligence*, and for accelerating *scientific discovery* by processing massive datasets in fields like genomics and climate science.
> > >
> > > *   **A Sustainable and Extensible Ecosystem:** To support these and future applications sustainably, we are refactoring Data-Juicer into a *modular architecture* (e.g., `datajuicer-core`, `datajuicer-mm`, `datajuicer-agent`, details can be found in our response to Reviewer cnBx). This modularity will lower the barrier for both users—who can install only what they need—and community contributors, empowering them to develop specialized extensions (e.g., a future `datajuicer-bio`) without needing to master the entire codebase. This strategic decision ensures the project's long-term health and adaptability to new domains.
> > >
> > > Thank you again for your constructive engagement. We respectfully hope this response can address your question and further strengthen your confidence in our work.

---

### Note · Authors · 2025-08-12

Dear Reviewers, AC and SAC,

Thank you for the reviewing effort and invaluable feedback on our submission. To assist in the assessment, we would like to provide a factual summary of the author-reviewer interactions, which may help provide a quick tour of the discussion status and facilitate a comprehensive evaluation.

Our paper began with initial ratings of [`cnBx`: 5, `Q2iW`: 5, `5StA`: 4, `m1C3`: 4]. The discussion period was instrumental in clarifying our work's contributions, and we are very grateful that it led two reviewers to explicitly state their intention to raise their scores:

*   Reviewer `5StA` (initial: 4) concluded: "*most of my concerns are addressed. I will raise my score to accept.*"
*   Reviewer `cnBx` (initial: 5) confirmed: "*... address my concerns ... Based on this I will update the score for the review.*"

We were further encouraged that Reviewer `m1C3` (initial: 4) affirmed that their "*main concerns are addressed.*" The reviewer prompted us to provide more extensive details on the system's broader applications and community adoption, which we hope can resolved their remaining questions.

Across all reviews, a clear consensus emerged acknowledging our work's key strengths. For example, reviewers recognized the system as "*comprehensive, completely implemented, and impressively mature*" (`Q2iW`), agreed that the problem is "*important*" and paper is "*well-written*" (`m1C3`), and noted that our claims are supported by "*detailed experiments on different scales*" (`5StA`). The discussions also enabled us to clarify key aspects and our plans for the final version:

1.  **Contribution & Track Alignment:** How our open-source infrastructure contribution directly aligns with broad NeurIPS community and the D&B track's call for papers (response to `5StA`).
2.  **Large-Scale Validation:** Confirmation that our system was evaluated on billion-scale multimodal data and has clear potential to larger-scale scenarios (response to `cnBx`).
3.  **Real-World Impact:** The system's broad adoption and use beyond a single organization (response to `m1C3`).
4.  **Narrative and Presentation:** Our action to foreground design rationale and improve figure clarity, addressing valuable feedback on presentation (response to `Q2iW`).

We are fully committed to incorporating all these feedback to deliver the highest quality manuscript.

Thank you again for your time and productive guidance!

Sincerely,
The Authors

---

### Decision · Program_Chairs · 2025-09-18

**Decision:**

Accept (spotlight)

**Comment:**

This paper introduces Data-Juicer 2.0, a comprehensive system featuring 100+ operators for processing text, image, video, and audio data, supporting tasks like analysis, synthesis, annotation, and model post-training. It has seamless integration for platforms like Hugging Face and Ray, and evaluations show its scalability, efficiently processing TB-level data with 10k+ CPU cores. It is open-sourced and already adopted in research and industry (Alibaba Cloud).

The reviewers all scored this submission positively, and all reviewers had their concerns addressed during the rebuttal. Some of the key strengths include that the benchmark is comprehensively implemented and tested, that the problem is important, and the paper generally well-written. Given the unanimously positive reviews, the fact that the submission includes a significant and novel open-source infrastructure contribution beyond just a benchmark, and the fact that it has already been validated through large-scale deployment in a system with billion-scale multimodal data, I believe the paper warrants recognition through spotlight presentation at neurips.